# Dynamic omnivory shapes the functional role of large carnivores under global change

Jörg Albrecht [1,2] ✉, Hervé Bocherens[3,4], Keith A. Hobson[5,22], Dorothée G. Drucker[3], Agnieszka Sergiel [2], Jon E. Swenson[6], Andreas Zedrosser[7,8], Adrian Marciszak[9], Elisabeth Iregren[10], Leena Drenzel[11], René Kyselý [12], Grzegorz Lipecki [13], Daniel Makowiecki[14], Jan Wagner [15], Tomasz Zwijacz-Kozica[16], Susanne A. Fritz[1,17,18,19], Eloy Revilla [20] & Nuria Selva [2,20,21]

Omnivory is increasingly recognized as a dynamic stabilizing force under environmental change. Despite its ubiquity across ecosystems, trophic levels and spatiotemporal scales, our empirical understanding of how omnivores respond to changing conditions in terrestrial ecosystems is limited. Here we combine macroecological and paleoecological approaches across seven bear species—the largest terrestrial carnivores—and discover they dynamically adapt their trophic position in food webs to resource availability. Throughout their ranges, bears shift to carnivory in unproductive ecosystems with short growing seasons and to herbivory in productive ecosystems with long growing seasons. In line with this, isotopic evidence from the Late Pleistocene and Holocene reveals a sharp decrease in the trophic position of the European brown bear in response to increasing net primary productivity and growing season length. These findings reveal a mechanism of trophic rewiring that alters the functional role of large carnivores in ecosystems and may simultaneously stabilize food web dynamics under global change.

Global change fundamentally reshapes the structure of terrestrial and aquatic food webs, which can have profound effects on entire ecosystems[1–3]. Numerous studies have investigated the mechanisms behind changes in the structure of food webs under global change[1]. However, an important mechanism that has received little attention is changes in trophic interactions due to the dynamic foraging behavior of consumers[4,5]. This mechanism is probably widespread and should be first detected in large omnivores at higher trophic levels, because they are adapted to rely on a wide range of resources, exhibit high behavioral flexibility, and often respond rapidly to environmental change[4,5]. As omnivores are common in food webs, a deterministic shift in the functional role of omnivores due to global change could result in the rewiring of entire food webs[4]. In addition, changes in the functional role of omnivores, for instance from predation to herbivory, have direct consequences for food-web dynamics[5] and key ecosystem functions such as nutrient cycling, energy flux, and biomass production[6]. Therefore, trophic responses of omnivores to global change could be a powerful indicator of critical transitions in food web structure and ecosystem functioning[4,5].

Global change may exert particularly strong effects on the foraging behavior of omnivores via changes in resource availability. For instance, land-use intensification strongly reduces the availability of net primary productivity (NPP) to wildlife via the conversion of natural vegetation to livestock and crop production systems[7]. On the other hand, anthropogenic nutrient inputs to ecosystems and food subsidies can increase the availability of resources to wildlife[8,9]. Moreover, prolonged growing seasons due to climate warming[10] can reduce seasonal bottlenecks in NPP and may release consumers from energetic constraints by reducing the metabolic costs of maintenance and foraging in warmer environments[11,12]. Accordingly, food web theory suggests that changes in resource availability and growing season length may cause shifts in the foraging behavior and trophic position of omnivores

('dynamic omnivory' hypothesis; Fig. 1)[4,5,12–14]. Yet, the trophic adaptation of large terrestrial omnivores to altered resource availability and climate remains poorly understood[4,5] because, traditionally, omnivory has been considered as a static trait[5] and studies that have investigated the dynamics of omnivory have mainly been conducted in micro- and mesocosm experiments or aquatic ecosystems[4,5,15,16]. In addition, most research has been conducted at local scales and over short periods, but little is known about the dynamics of omnivory at larger spatio-temporal scales[4,5].

Here, we aim to fill these knowledge gaps by combining macroecological and paleoecological approaches to investigate how large terrestrial omnivores adapt their trophic position in food webs to changing NPP and growing season length at global and millennial scales. We leverage the unique insights gained by both approaches[17,18] to infer how large terrestrial omnivores adapt their trophic position in food webs to global change. We focus on the seven extant terrestrial bear species (Order: Carnivora, Family: Ursidae), which are the largest terrestrial omnivores and occupy a wide range of biomes from the arctic tundra to tropical rainforests. Unlike most other large carnivores, bears show a preference for low-protein diets and have relatively weak craniodental adaptations to carnivory[19–22], which allows them to maintain a high degree of dietary flexibility. Owing to their broad dietary niches, bears contribute to a multitude of ecosystem processes, such as predation, scavenging, frugivory, and herbivory that can have strong impacts on prey populations[23], plant regeneration[24,25], nutrient cycling[26,27] and energy fluxes[6] within and across terrestrial and aquatic ecosystems.

## Results

### Macroecological analysis

We first investigated how the trophic position of extant bears is related to NPP and growing season length across their geographic ranges (Fig. 2). To do so, we compiled a comprehensive database of dietary compositions based on micro-histological analyses of fecal and stomach contents throughout the geographic ranges of the seven extant terrestrial bear species, using 210 records from 155 studies (Fig. 2a)[28]. We used annual diets, which integrate foraging behavior across the entire active season of the focal species, to enable meaningful comparisons across study locations. We excluded the polar bear

(*U. maritimus*) from our analysis, because this species almost exclusively hunts for marine prey on Arctic Sea ice, but fasts on land during the ice-free season[29]. Therefore, the minor contribution of terrestrial food sources during the ice-free season is not representative of the species' trophic niche[30–32]. Based on the dietary data for the remaining seven terrestrial bear species, we used a Bayesian hierarchical model to estimate the trophic position as the percentage of the dietary energy that is contributed by animal prey (including vertebrates and invertebrates), while accounting for the effects of digestibility and energy content of different food sources (Fig. 2b; *Methods*; Supplementary Table 1). In a second step, the model related the estimated trophic position to NPP (kg C $m^{-2}$ $a^{-1}$) and meteorological growing season length[33] (months with mean temperature $T > 0$ °C; hereafter growing season length) at each of the study locations (Fig. 2c, d). The temperature threshold of 0 °C demarcates the growth period of frost-resistant plants[34], as well as the active period of bear populations in temperate and boreal biomes (e.g., *Ursus arctos* and *U. americanus*)[35,36]. To account for the potential effects of competition between sympatric bear species, we also included a factor indicating whether a focal population was located within the range of another larger (dominant) or smaller (subordinate) bear species[37,38]. The model included random factors for study and species identity to account for the non-independence of data from the same species and study and propagated all uncertainties associated with the estimated trophic position through the entire model.

The model revealed that the trophic position of extant bears across their geographic ranges is negatively related to NPP (−0.18 [−0.30, −0.066], $p_d = 0.99$; median [90% Equal-Tailed Interval], and posterior probability of the relationship being negative) and growing season length (−0.41 [−0.57, −0.26], $p_d = 1$; Fig. 2c, d; Supplementary Table 2). Thus, terrestrial bears generally occupy higher trophic positions in ecosystems with low productivity and short growing seasons and lower trophic positions in ecosystems with high productivity and long growing seasons. A decline in trophic position may be attributable to (i) a population effect, where local bear populations alter their foraging behavior relative to NPP and growing season length, or (ii) a species occurrence effect, where bear species with a certain trophic position simply do not occur in environments with a particular NPP and growing season length. To disentangle these two effects, we ran an

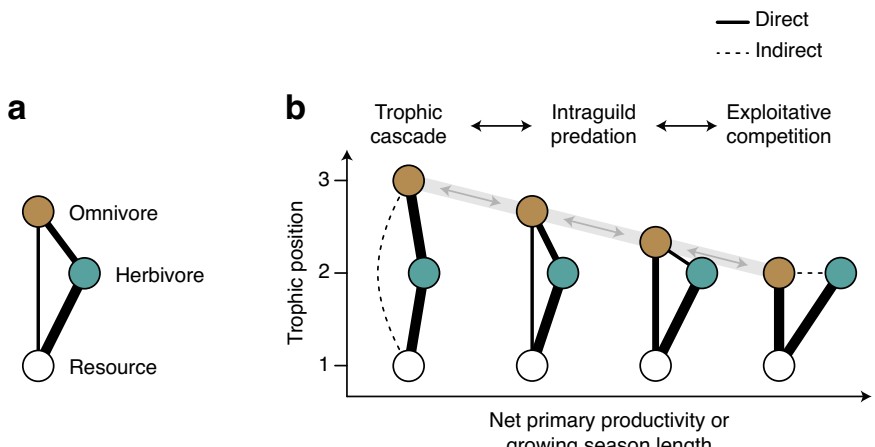

**Fig. 1 | The trophic adaptation of omnivores to resource availability and growing season length alters the structure of tritrophic food-web motifs.**
**a** Schematic of a simple tritrophic food web with omnivory containing a basal resource, an herbivore and an omnivorous top predator. **b** The 'dynamic omnivory' hypothesis from food web theory[5] suggests that with increasing resource availability at the base of food webs the structure of tritrophic food webs shifts from a trophic cascade, where the top predator feeds exclusively on lower-level consumers, to various levels of intraguild predation, where the top predator feeds on both consumers and resources, to exploitative competition, where the top predator feeds exclusively on the resource and competes with the lower-level consumer for this resource. Note the shift in the relative strengths of the omnivore–resource interaction and the omnivore–herbivore interaction in response to net primary productivity and growing season length.

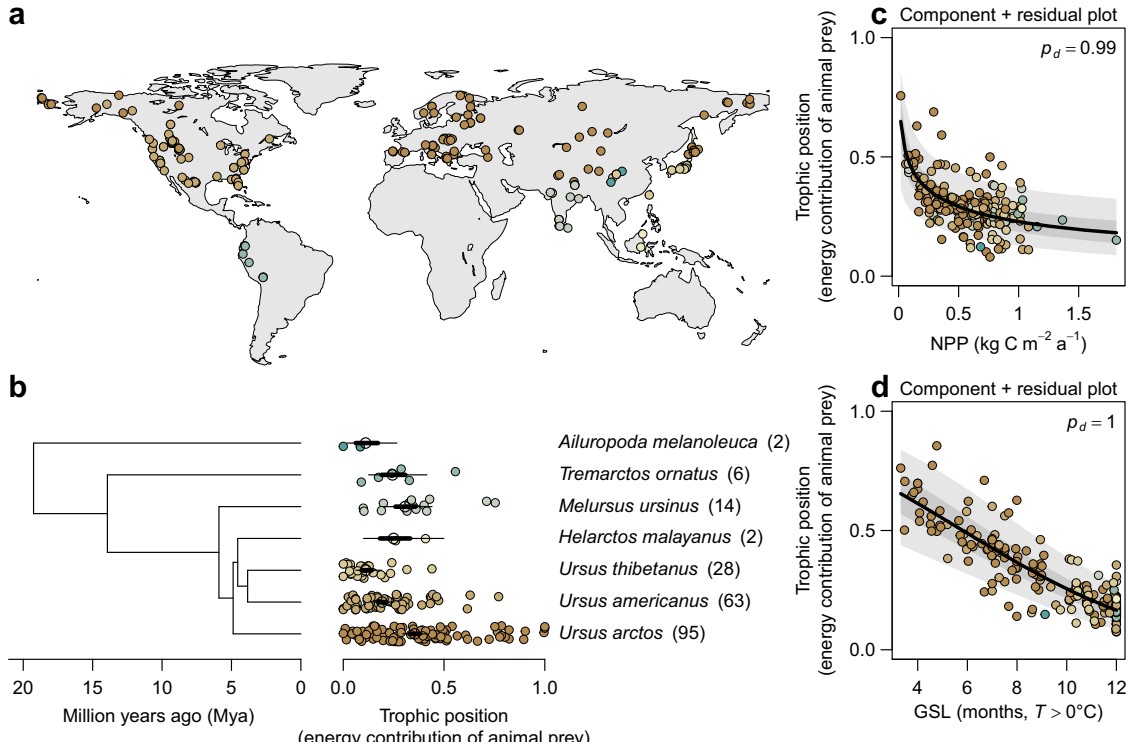

**Fig. 2 | Global variation in the trophic position of bears. a** Geographic distribution of diet studies based on micro histological analyses of fecal and stomach content included in the meta-analysis. The background map is based on public data from Natural Earth (www.naturalearthdata.com). **b** Phylogenetic tree of the lineage of bears[102] (excluding *U. maritimus*) alongside species-specific trophic position, estimated as the relative dietary energy contribution of animal prey. The phylogenetic tree is plotted for illustrative purposes only. Black circles, thick, and thin lines represent posterior estimates (median, 50% and 90% ETIs [Equal-Tailed Intervals]) of species-specific trophic position, whereas small circles represent the estimated trophic position in each study location. Numbers in brackets give the sample size for each species. **c, d** Component + residual plots showing the partial relationships of trophic position with (**c**) net primary productivity (NPP, kg C m$^{-2}$ a$^{-1}$) and (**d**) meteorological growing season length (GSL, months with $T > 0$ °C) while conditioning on the effects of the other predictors. Black lines, dark and light gray bands represent estimated relationships and uncertainty (median, 50% and 90% ETIs, respectively). Sample sizes are $n_{observations} = 210$, $n_{study} = 155$, $n_{species} = 7$. $p_d$ is the posterior probability of a relationship being the same direction as the median.

additional model where we separated the effects of NPP and growing season length into these two components (*Methods*). The two-component model showed that variation in trophic position was exclusively explained by population effects, and not by the species occurrence effect (Supplementary Table 2). These effects were robust despite additional effects of competition with sympatric bear species, which caused a decrease in the trophic position of subordinate bear species (−0.50 [−0.84, −0.16], $p_d = 0.99$) but no change in the trophic position of dominant bear species (0.059 [−0.26, 0.37], $p_d = 0.62$; Supplementary Table 2). This indicates that the trophic position of extant bears is a flexible trait that emerges from adaptations to local resource availability, growing season length, and interspecific competition with other sympatric bear species at the population level.

## Paleoecological analysis

In a second step, we investigated how European brown bears adapted their trophic position to the marked increases in NPP and growing season length at the transition from the Late Pleistocene to the Holocene (Fig. 3a). To do so, we used stable isotope analyses of collagen extracted from a comprehensive collection of fossil and subfossil bone and tooth remains of brown bears (*n* = 219) and red deer (*Cervus elaphus, n* = 372) across Europe, covering the last 55 ka BP (Fig. 3b)[28]. The low δ¹³C and δ¹⁵N values (‰) of brown bear collagen allowed us to rule out the consumption of marine resources (e.g., anadromous fish; see "*Methods*"). Therefore, we estimated the trophic position of brown bears based on δ¹⁵N of collagen through time using a Bayesian hierarchical model, in which the δ¹⁵N values of red deer from the same

region were included as a dietary baseline for a strict herbivore (Fig. 3c; *Methods*). The model accounted for the effects of altitude[39] and the type of sampled material (bones or teeth)[38] on δ¹⁵N values of brown bears and red deer and propagated all uncertainties associated with these factors, as well as with trophic fractionation[38,40] through the entire model (*Methods*; Supplementary Table 3). In the last step, the model estimated the relationship of the trophic position of brown bears with NPP and growing season length, which were derived from global climate and vegetation models (HadCM3 and BIOME4)[41] (Fig. 3a).

We found that European brown bears occupied higher trophic positions during the Late Pleistocene than during the Holocene (Fig. 3c). This observed decrease in trophic position of European brown bears from the Late Pleistocene to the Holocene was closely associated with increasing NPP (−0.17 [−0.30, −0.029], $p_d = 0.97$) and growing season length (−0.19 [−0.33, −0.041], $p_d = 0.98$) at the onset of the Holocene (Fig. 3d, e; Supplementary Table 3). Consequently, the occupied trophic positions were highest during the period characterized by the lowest NPP and shortest growing seasons (period GS3 to GI8 in Fig. 3c) and lowest during those periods characterized by high NPP and long growing seasons (periods Greenlandian and North-grippian in Fig. 3c). Most strikingly, the paleoecological patterns for brown bears closely resembled the macroecological patterns observed across the geographic ranges of the seven extant terrestrial bear species (Figs. 2 and 3). These results are in strong agreement with the 'dynamic omnivory' hypothesis, predicting that omnivores adapt their trophic position to resource availability and growing season length (Fig. 1).

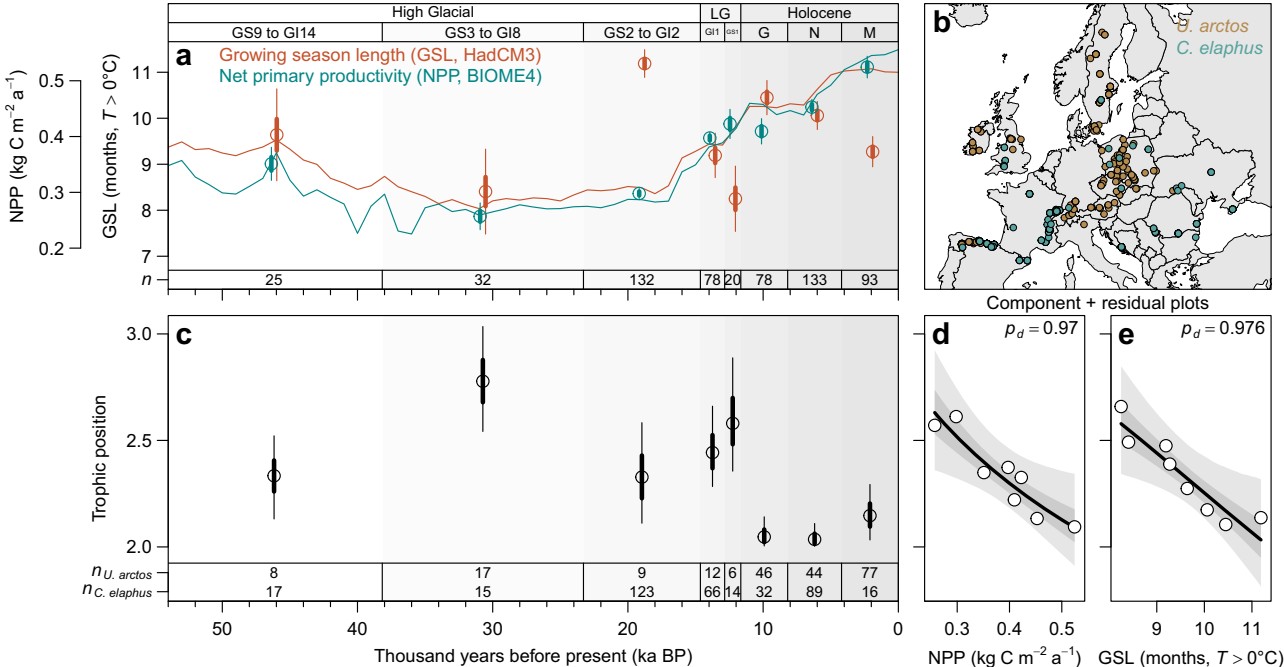

**Fig. 3 | Change in trophic position of the brown bear (*Ursus arctos*) from Late Pleistocene to Holocene. a** Net primary productivity (NPP, kg C m$^{-2}$ a$^{-1}$) and meteorological growing season length (GSL, months with $T > 0$ °C), based on global climate and vegetation models (HadCM3 and BIOME4)[41] during the last 55,000 years. Circles, thick and thin error bars represent mean NPP and growing season length, as well as 50% and 90% CIs [Confidence Intervals], across the subfossil bone and tooth samples of brown bear and red deer (*Cervus elaphus*) in each time bin (*Methods*). Lines indicate mean temporal trends in NPP and growing season length across all sample locations shown in (**b**). Abbreviations are LG, Late Glacial; GS9 to GI14, Greenland Stadial 9 to Greenland Interstadial 14; GS3 to GI8, Greenland Stadial 3 to Greenland Interstadial 8; GS2 to GI2, Greenland Stadial 2 to Greenland Interstadial 2; GI1, Greenland Interstadial 1; GS1, Greenland Stadial 1; M, Meghalayan; N, Northgrippian; G, Greenlandian. Sample sizes (*n*) are given at the bottom of the panel. **b** Locations of subfossil bone and tooth samples of brown bear and red deer across Europe. The background map is based on public data from Natural Earth (www.naturalearthdata.com). **c** Trophic position of the brown bear as estimated by a Bayesian hierarchical model with red deer as baseline, where a value of 2 corresponds to a strict herbivore (i.e., trophic level = 2) and values larger than 2 reflect an increasing dietary contribution of animal prey. Black circles, thick and thin lines represent posterior estimates (median, 50% and 90% ETIs, respectively). Sample sizes for brown bear ($n_{U. arctos}$) and red deer ($n_{C. elaphus}$) are given at the bottom of the panel. **d**, **e** Component + residual plots showing the partial relationships of trophic position with NPP and growing season length, conditioned on the effect of the other predictor variable. Black lines, dark and light gray bands represent estimated relationships and uncertainty (median, 50% and 90% ETIs, respectively). $p_d$ is the posterior probability of a relationship being the same direction as the median.

## Discussion

The observed trophic adaptation to NPP and growing season length is in line with predictions from food web models and with empirical observations from micro- and mesocosm experiments and lake ecosystems[4,5,12–16]. We attribute the observed trophic adaptation to three nonexclusive mechanisms: (i) tracking seasonal peaks in plant productivity (e.g., fruit and seed production), facilitated by the mobility of these large omnivores[4,5]; (ii) shifts in foraging preferences if foraging for plant resources is energetically more efficient than hunting or scavenging, due to lower search and handling times in dense vegetation[27,42]; and (iii) preferential foraging for plant resources in warmer environments, where reduced metabolic costs of maintenance and foraging alleviate energetic constraints[11,12]. These results are supported by previous work reporting trophic adaptations to seasonal and annual variability in resource availability within populations of six of the seven terrestrial bear species[19–21,27,43–45]. The only exceptions among ursids in this context are the polar bear and the giant panda (*Ailuropoda melanoleuca*), which represent the extremes of the carnivory–omnivory–herbivory gradient in our study (Fig. 1): Polar bears are almost strictly carnivorous, preying on a range of marine mammals on sea ice[46] and consuming plant resources (e.g., berries) only in southern areas where they are forced ashore during the ice-free period in summer[30,44,47]. At the other extreme, giant pandas are strictly herbivorous, consuming almost exclusively bamboo[48,49] and only occasionally scavenging or predating on small mammals[20]. Therefore, we expect that only those species that are omnivorous and have been shown to switch seasonally between animal- and plant-based diets

depending on availability are able to adapt their trophic position to changes in resource availability. Apart from the above limitation, our study highlights that, similar to aquatic food webs, the trophic interactions of large omnivores in terrestrial food webs are not static but can change dynamically in response to altered resource availability. The fact that extant terrestrial bears show a consistent trophic response to resource availability across their geographic ranges puts the findings of these previous studies into a macroecological perspective. Over long time scales, the observed trophic adaptation to resource availability throughout the geographic ranges of bears might even be a driver of intraspecific genetic differentiation and population structure[50,51].

The sharp decrease in the trophic position of European brown bears at the onset of the Holocene (11.7 ka BP) challenges the hypothesis that the trophic position of the European brown bear decreased due to competitive release after the extinction of the cave bear (*Ursus spelaeus*; 24.3–26.1 ka BP)[38,52]. Our results rather suggest that European brown bears primarily adapted their trophic niche to enhanced resource availability, whereas competitive release seemingly played a secondary role. This is supported by the fact that we found consistent effects of NPP and growing season length across extant terrestrial bears after accounting for the potential effects of competition between sympatric species (Supplementary Table 2). The relatively steady trophic position of European brown bears throughout the Early- and Mid-Holocene highlights that trophic adaptations can develop into stable trophic strategies that persist within populations when environmental conditions remain constant (Fig. 3c). Overall, the

strong agreement between the macroecological and paleoecological approaches suggests that the trophic position occupied by extant bears in food webs across their current geographic ranges is the result of trophic adaptations to local resource availability and climate.

The shift in the trophic position of brown bears in response to environmental change at the onset of the Holocene (Fig. 3) indicates that current global change may have profound effects on the trophic position and functional role of omnivores in terrestrial food webs. For instance, land-use intensification reduces the availability of primary production to wildlife[7] and may cause opportunistic shifts in foraging behavior to alternative food sources, including livestock or crops[53]. In addition, anthropogenic nutrient inputs to ecosystems and food subsidies increase the availability of resources, which can also affect the foraging behavior and trophic interactions of omnivores[9]. Finally, the increasing length of growing seasons associated with climate warming[10] may release omnivores from seasonal bottlenecks in primary production and physiological constraints[11,12], which may cause dietary shifts from animal prey to plant resources[5,54]. The release from seasonal bottlenecks in NPP associated with warmer autumn and spring temperatures also affects the activity and wintering behavior of bears[35,36], which might in part explain shortened hibernation periods and cases of non-hibernation in some bear populations[55].

Despite the signal of trophic adaptation of populations to environmental change in our study, it remains unclear to what extent species that show morphological adaptations to specific diet types (e.g., herbivory in the giant panda)[19–21,56] or populations that occur at the physiological or ecological limits of their species' range in extreme environments (e.g., arctic, desert, or alpine environments) are able to cope with future changes in environmental conditions. The adaptive capacity of these species and populations will depend on the magnitude and velocity of environmental change, as well as on whether alternative food sources meet their nutritional requirements[4,57]. For instance, it has been suggested that the switch of polar bears from marine prey to terrestrial food sources (e.g., berries or bird nests) in response to reductions in sea-ice extent is associated with declines in body size, body condition, reproduction, and population sizes[58]. Similarly, due to their specialization on bamboo, the capacity of giant pandas to adjust their trophic niche to variations in resource availability is likely limited[48]. Investigating how the interplay of behavioral, morphological, and physiological factors constrains or facilitates the trophic flexibility of omnivores may provide further insights into their adaptive capacity to environmental change.

A growing body of literature highlights the pervasive impact of large animals on the structure and functioning of Pleistocene ecosystems and the cascading effects of historic declines and losses of this megafauna on ecosystems[59]. Our study adds another dimension to these studies because the observed trophic adaptation of large terrestrial omnivores to altered resource availability under global change could have impacts on food webs and ecosystem functions that go beyond the effects of population declines. On the one hand, food-web models indicate that rapid responses of omnivores to changing resource availability stabilize population dynamics under global change[5]. Despite these potentially stabilizing effects, changes in the trophic position of large omnivores directly affect the structure of food webs (e.g., food chain length and interaction strengths) and have strong effects on nutrient cycling, energy fluxes, and biomass stocks in ecosystems[6]. Apart from these direct impacts on food webs and ecosystem functions, the trophic adaptation of omnivores may also translate into a higher frequency of human–wildlife conflicts, if omnivores increasingly use anthropogenic resources in agricultural landscapes[53].

By combining macroecological and paleoecological approaches, our study provides strong empirical evidence that large terrestrial omnivores adapt their trophic position in food webs dynamically to resource availability and climate, with a general decrease in trophic

position in response to higher primary production and longer growing seasons. Our findings complement empirical observations from previous small-scale experiments and aquatic ecosystems, suggesting that the trophic adaptation of omnivores to resource availability is common across aquatic and terrestrial ecosystems. Because the trophic position of omnivores in food webs is closely linked to their functional role in ecosystems, their trophic adaptation to current global change is likely to have cascading effects on the structure of food webs and ecosystem functions.

## Methods

### Macroecological analysis

**Trophic data.** To obtain dietary data for the seven extant terrestrial bear species, we conducted a literature search in the Web of Science Core Collection for publications until 2018. We searched titles and abstracts using the keywords "(food OR feed* OR diet* OR forag* OR nutri* OR scat* OR fec* OR faec* OR stomach* OR gut*) AND (ursidae OR ursus OR melursus OR tremarctos OR helarctos OR ailuropoda OR bear* OR panda)". We also checked the reference lists of the retrieved publications for additional studies that had not been identified by the keyword search. To be included in our analysis, the studies had to meet the following criteria: (i) Diet assessment was based on microhistological analyses of food remains in scats or stomachs. (ii) Sample collection in the field covered the entire seasonal activity period of the focal species. (iii) All food items, instead of only a subset, were reported. (iv) Reported food items could be mapped onto established macroecological diet classification schemes[60]. (v) The number of collected samples was reported. (vi) Sufficient data to calculate one of the following measures was reported: relative frequency of occurrence of food items ($F_i$), calculated as the number of occurrences of food item $i$ divided by the total number of occurrences of all food items ($F_i = f_i / \sum_{i=1}^{I} f_i$); the relative volume of food items ($V_i$), calculated as the mean volume of food item $i$ in scats or stomachs divided by the total volume of all food items, that is ($V_i = v_i / \sum_{i=1}^{I} v_i$); the relative dietary contribution based on ingested dry weight of food items ($D_i$), calculated as the product of the relative volume of food item $i$ and correction factors $c_{Di}$ that account for differences in the digestibility of food items and normalizing to proportions ($D_i = c_{Di} V_i / \sum_i c_{Di} V_i$); or relative dietary energy contribution of food items ($E_i$), calculated as the product of the relative dietary dry weight contribution of food item $i$ and correction factors $c_{Ei}$ that account for differences in the energy content of food items and normalizing to proportions ($E_i = c_{Ei} D_i / \sum_i c_{Ei} D_i$). We excluded the polar bear from our analysis, because this species almost exclusively hunts for marine prey on Arctic Sea ice, but fasts on land during the ice-free season[29]. Therefore, the minor contribution of terrestrial food sources during the ice-free season is not representative of the species' trophic niche[30–32].

In total, 155 publications met the above criteria, including articles in scientific journals, master's and PhD theses, and gray literature (e.g., non-commercial publications in the form of technical reports or conference proceedings)[28]. For each publication, we extracted the geographic coordinates of the study location, the type of collected samples (i.e., scats or the stomachs of dead individuals), the number of samples, and the reported food items in scats or stomachs. We mapped the reported food items onto one of 20 dietary categories based on an established macroecological diet classification scheme[60] (Supplementary Table 1) and recorded the quantity of food items in the diet using the above described variables. To convert estimates of relative volume into relative dietary contribution or dietary energy contribution, we compiled correction factors that account for differences in the digestibility and energy content of food items from the literature (Supplementary Table 1).

**Environmental variables.** For the geographic location of each record, we extracted data on net primary productivity (NPP, kg C m$^{-2}$ a$^{-1}$)

(Terra MODIS17A3, spatial resolution: 0.5°)[61] and near-surface daily average air temperature at monthly resolution (CHELSA, spatial resolution: 0.5°)[62]. We calculated growing season length as the number of months, in which the mean temperature was above 0 °C. This temperature threshold demarcates the thermal growth period of frost-resistant plants[34], as well as the active period of bear populations in temperate and boreal biomes (e.g., *Ursus arctos* and *U. americanus*, respectively)[35,36]. NPP and growing season length were moderately positively correlated (Pearson's $r = 0.61$, $t = 11.2$, df = 208, $P < 0.001$, Supplementary Fig. 1), and variance inflation factors indicated that the correlation between both predictors did not affect estimates of effect sizes (see section *Model implementation and assessment of model fit*).

**Species co-occurrence.** We quantified species co-occurrences using IUCN species distribution maps[63]. For the geographic records of each species, we determined whether a location fell within the geographic range of one or more other bear species. Then we created a categorical variable with three levels (no co-occurrence, subordinate, and dominant), indicating whether the focal species at a given location co-occurred with another bear species and, if so, whether it was smaller (subordinate) or larger (dominant) than the co-occurring species.

We did not account for co-occurrence with other large carnivores in the analysis (e.g., felids or canids), as bears have evolved from carnivores with high-protein diets to omnivores that consume low-protein diets, thus, reducing interspecific competition with other large carnivores[22]. Moreover, bears are well-documented kleptoparasites, often monopolizing carrion resources and significantly limiting carrion access and predatory behavior in other large carnivores (e.g., wolves, cougars, and pumas)[64–70]. In interspecific encounters with other carnivores, bears typically dominate due to their large body size or are able to defend themselves by charging, as observed in encounters with tigers (*Panthera tigris*)[71–73]. In cases of interspecific killing, bears are typically the killer species, while rare instances of bears being killed usually involve denning adults, cubs of the year, or aggressive encounters with tigers in Indian tiger reserves and the Russian Far East[71–76].

**Macroecological model.** To analyze global patterns of trophic position across the seven extant terrestrial bear species, we used a Bayesian hierarchical model that consisted of two sub-models. Sub-model I was used to (i) represent the relative volume of food items in sampled materials for those datasets with missing information on relative volume (107 of 210 datasets), based on the close relationship with the relative frequency of occurrence of food items (Supplementary Fig. 2) and (ii) to estimate the relative dietary energy contribution of animal prey, including vertebrates and invertebrates (i.e., the trophic position). All uncertainties associated with the data imputation and estimation of trophic position were propagated through the entire model. Sub-model II was then used to estimate the relationship of the trophic position with NPP, growing season length, as well as co-occurrence with other subordinate or dominant bear species.

To inform sub-model I, we first quantified the relationship between the relative frequency of occurrence ($F$) and relative volume ($V$) for those datasets (i.e., unique study-year or study-location combinations), in which both variables had been reported ($n = 106$ datasets). For each dataset $j$, we quantified the relationship between $F$ and $V$ (both log-transformed) using geometric mean regression to account for the fact that both variables are subject to measurement error[77]. We calculated the geometric mean slope as $\beta_{j,k=2} = \text{sign}\left(\sigma_{FV_j}\right)\sqrt{\sigma^2_{V_j}/\sigma^2_{F_j}}$, and the associated intercept as $\beta_{j,k=1} = \bar{V}_j - \bar{F}_j\beta_{j,k=2}$, where $\bar{V}_j$ and $\bar{F}_j$ are the mean values of both variables for dataset $j$. The distribution of geometric mean slopes and associated intercepts across the datasets was then used to inform the data imputation in sub-model I. In particular, sub-model I assumed

that the slopes and intercepts follow a multivariate normal distribution around $\mu_{\beta_j}$ and variance-covariance matrix $\Sigma_\beta$:

$$\beta_{jk} \sim \text{Normal}\left(\mu_{\beta_{jk}}, \Sigma_\beta\right) \tag{1}$$

$$\mu_{\beta_{jk}} \sim \text{Normal}\left(0, 10^{-4}\right) \tag{2}$$

$$\Sigma_\beta \sim \text{scaled} - \text{inverse} - \text{Wishart}(\mathbf{s}, \text{df}) \tag{3}$$

where $\Sigma_\beta$ is a $2 \times 2$ variance–covariance matrix. We used a non-informative scaled inverse Wishart prior[78] with scale $s_k = 1$ and d.f. = 2. This prior follows a half-$t$ distribution for the standard deviation, $\sigma_{kk} \sim st^+_{\text{d.f.}}$, and has a marginal uniform prior distribution for the correlation parameter $\rho_{kl}$. For each dataset $j$ without information on $V_{i[j]}$ the model then drew an intercept and slope from the multivariate distribution and estimated values of $V_{i[j]}$ using the observed values of $F_{i[j]}$ based on the following regression formula:

$$\mu_{V_{i[j]}} = \beta_{j,k=1} + \beta_{j,k=2}F_{i[j]} \tag{4}$$

$$V_{i[j]} \sim \text{Normal}\left(\mu_{V_{i[j]}}, \sigma^2_V\right) \tag{5}$$

$$\sigma^2_V \sim \text{scaled inverse} - \text{Gamma}(s, \text{df}) \tag{6}$$

We modeled $V_{i[j]}$ using a normal distribution with mean $\mu_{V_{i[j]}}$ and residual variance $\sigma^2_V$. For the residual variance, we used a weakly informative scaled inverse gamma prior[79] with scale $s = 1$ and two degrees of freedom (d.f. = 2), which for the standard deviation follows a half-$t$ distribution. Then the relative dietary energy contribution $E$ of each food item was calculated for each dataset $j$ by multiplying the values of $V$ with correction factors $c_D$ and $c_E$ that account for differences in the digestibility and energy content of food items and normalizing to proportions:

$$E_{i[j]} = c_{D_{i[j]}} c_{E_{i[j]}} V_{i[j]} / \sum\nolimits_{i[j]} c_{D_{i[j]}} c_{E_{i[j]}} V_{i[j]} \tag{7}$$

Based on the relative dietary energy contribution $E$ of each food item, we calculated for each dataset $j$ the trophic position as the relative dietary energy contribution of animal prey $P_j$ (including vertebrates and invertebrates). The process model then quantified the relationship of $P$ with NPP, growing season length, as well as co-occurrence with other subordinate or dominant bear species. The model can be written as follows:

$$\mu_j = \sum\nolimits_k \beta_k \mathbf{X}_{jk} + \gamma_{l[i]} + \delta_{m[i]} \tag{8}$$

with

$$y_j \sim \text{Normal}\left(\mu_j, \sigma^2_\varepsilon\right) \tag{9}$$

$$\gamma_l \sim \text{Normal}\left(0, \sigma^2_\gamma\right) \tag{10}$$

$$\delta_m \sim \text{Normal}\left(0, \sigma^2_\delta\right) \tag{11}$$

$$\sigma^2_\varepsilon, \sigma^2_\gamma, \sigma^2_\delta \sim \text{scaled inverse} - \text{Gamma}(s, \text{df}) \tag{12}$$

$$\beta_{jk} \sim \text{Normal}\left(0, 10^{-4}\right) \tag{13}$$

where $\mu_j$ is the expected value of the $j$th observation, **X** is a model matrix including an intercept and the explanatory variables (NPP [$\log_{10}$-transformed], growing season length, and co-occurrence with other subordinate or dominant bear species [dummy-coded]) and an associated vector of parameters $\beta$, and $\gamma_l$ is a random effect for the $l$th study, which is normally distributed around 0 with variance $\sigma_\gamma^2$, and $\delta_m$ is a random effect for the $m$th species, which is normally distributed around 0 with variance $\sigma_\delta^2$. We modeled the trophic position with a normal distribution around $\mu_j$ and residual variance $\sigma_\varepsilon^2$. For modeling purposes, we first compressed values of $P_j$ which were in the closed interval $(0 \le y \le 1)$ into the open interval $(0 < y < 1)$ and subsequently applied a probit transformation: $y_j = \text{probit}((P_j(n-1) + 0.5)/n)$, in which $n$ is the total number of observations[80]. The compression was necessary, because the probit function is defined on the open interval $(0 < y < 1$; range of $P_j$ after transformation: $0.00034$–$0.99934$). We used an uninformative normal prior with a mean of 0 and a variance of $10^4$ for the fixed effects and weakly informative scaled inverse gamma priors with scale $s = 1$ and two degrees of freedom (d.f. = 2) for variances of random effects and residuals. To separate *population effects*, where local bear populations alter their foraging behavior relative to NPP and growing season length, from *species occurrence effects*, where bear species with a certain trophic position simply do not occur in environments with a particular NPP and growing season length, we ran an additional two-component model in which we applied group mean centering to these two predictor variables (Supplementary Table 2).

## Paleoecological analysis

**Database of subfossil and fossil remains.** To reconstruct the trophic position of the European brown bear during the Late Pleistocene and Holocene, we compiled a database of 591 dated and georeferenced subfossil and fossil remains of brown bears and red deer using published sources ($n = 483$ samples) and material from museum collections ($n = 108$; see *Data Availability* Statement)[28]. Dates of specimens were based on direct radiocarbon ($^{14}$C) dating ($n = 237$) or based on the context of the excavation sites ($n = 354$). Contextual dates were either obtained based on related artifacts ($n = 196$) or in case of three sites (El Miron, Kiputz IX, and Emine-Bair-Khosar) based on age-depth models ($n = 158$). Radiocarbon dating of material from museum collections ($n = 55$) was done using accelerator mass spectrometry at the Poznań Radiocarbon Laboratory (Poz), Poznań, Poland. Dated fossils without geolocations were geocoded manually using the name of the fossil site. The quality and reliability of all radiocarbon dates was assessed based on dating method, stratigraphy, association and material[81]. This resulted in 219 samples of brown bear and 372 samples of red deer. Age-depth models were fitted using package *Bchron* (version 4.7.6)[82] in *R* (version 4.4.2)[83]. The $^{14}$C ages of these fossils were calibrated using package *Bchron* and the IntCal13 curve[84].

**Stable isotope analysis.** Collagen was extracted from bone or tooth of subfossil and fossil specimens following previously established protocols[85,86]. The extraction process included a step of soaking in 0.125 M NaOH between the demineralization and solubilization steps to achieve the elimination of lipids and humic acids. Elemental analysis ($C_{coll}$, $N_{coll}$) and isotopic analysis ($\delta^{13}C_{coll}$, $\delta^{15}N_{coll}$) were conducted at two laboratories. At the Department of Geosciences of Tübingen University, Germany, an NC2500 CHN-elemental analyzer was coupled to a Thermo Quest Delta+ XL mass spectrometer. At the Environment Canada Stable Isotope Laboratory in Saskatoon, Canada, collagen was combusted at 1030 °C in a Carlo Erba NA1500 or Eurovector 3000 elemental analyser coupled with an Elementar Isoprime or a Nu Instruments Horizon isotope ratio mass spectrometer. The international standard for $\delta^{13}C$ measurements was the Vienna Pee Dee Belemnite (VPDB) carbonate that for $\delta^{15}N$ atmospheric nitrogen (AIR). Analytical error, based on within-run replicate measurement of

laboratory standards (Saskatoon: BWBIII keratin and PRCgel; Tübigen: albumen, modern collagen, USGS 24, IAEA 305 A), was ±0.1‰ for $\delta^{13}C$ values and ±0.2‰ for $\delta^{15}N$ values. Reliability of the $\delta^{13}C_{coll}$ and $\delta^{15}N_{coll}$ values was established by measuring its chemical composition, with $C/N_{coll}$ atomic ratio ranging from 2.9 to 3.6 ref.[87]., and percentage of $C_{coll}$ and $N_{coll}$ above 8% and 3%ref.[88]., respectively.

**Trophic discrimination factors and corrections for material type and elevation.** Because of the general isotopic enrichment in nitrogen between trophic levels, consumers exhibit higher $\delta^{15}N$ values than their food resources[38]. We accounted for this effect in the analyses using data on trophic discrimination factors based on bone collagen of predators and their prey from the literature[40].

Nitrogen isotope values ($\delta^{15}N$) generally decrease with increasing altitude[39]. We accounted for this effect of altitude in the analyses by correcting the raw $\delta^{15}N$ values using reference data from the literature[39] to model the relationship between elevation and the nitrogen isotope composition of plants and herbivores.

In long-lived large mammals, adult bone collagen reflects the average isotopic composition of the diet over several years preceding death, whereas collagen in tooth dentine primarily accumulates during the period of tooth development[38]. Since brown bears possess brachydont teeth dental growth ceases early in life, during a stage when a substantial proportion of nutrients is still derived from lactation[38]. Consequently, collagen extracted from dentine may exhibit higher $\delta^{15}N$ values than bone collagen from the same individual[38]. We accounted for this effect in the analyses by correcting the isotope ratio of teeth using data from the literature[38] on the difference in $\delta^{15}N$ values between collagen from tooth and bone material of the same individuals.

**Potential consumption of marine resources by brown bears.** To rule out the consumption of marine resources (e.g., anadromous fish) by brown bears, we compared the $\delta^{13}C$ and $\delta^{15}N$ values of the brown bear material with those of a range of marine mammal species from the literature[89] (Supplementary Fig. 3). The $\delta^{13}C$ and $\delta^{15}N$ values of brown bear material were much lower than those of material from marine mammals, but very similar to those of the red deer material (Supplementary Fig. 3), indicating negligible consumption of marine resources by brown bears in our data. Therefore, we estimated the trophic position of brown bears based on $\delta^{15}N$ of collagen through time using the $\delta^{15}N$ values of red deer from the same region as a dietary baseline for a strict herbivore.

**Environmental variables.** We used package *pastclim* (version 2.2.0)[90] to annotate each sample with spatiotemporally explicit data on net primary productivity (NPP, kg C m$^{-2}$ a$^{-1}$) from a global dynamic vegetation model (BIOME4, spatial resolution: 0.5°)[41], near-surface daily average air temperature at monthly resolution from a global climate model (HadCM3, spatial resolution: 0.5°)[41], and with data on elevation from a global elevation model (GMTED 2010, spatial resolution: 0.0083°)[91]. Analogous to the macroecological analysis, we calculated growing season length as the number of months in which the near-surface daily average air temperature was above 0 °C. NPP and growing season length were only very weakly correlated (Pearson's $r = -0.06$, $t = -0.14$, df = 6, $P = 0.89$, Supplementary Fig. 4), and variance inflation factors indicated that the correlation between both predictors did not affect estimates of effect sizes (see section *Model implementation and assessment of model fit*).

**Temporal resolution of analysis.** For the analysis, we defined eight time periods that were used to bin the data (Fig. 3): Meghalayan (0–4.2 ka BP), Northgrippian (4.2–8.2 ka BP), Greenlandian (8.2–11.7 ka BP), Greenland Stadial 1 (GS-1, 11.7–12.8 ka BP), Greenland Interstadial 1 (GI-1, 12.8–14.6 ka BP), Greenland Stadial 2 to Greenland Interstadial 2 (GS-2 to GI-2, 14.6–23.3 ka BP), Greenland Stadial 3 to Greenland

Interstadial 8 (GS-3 to GI-8, 23.3–38.2 ka BP), Greenland Stadial 9 to Greenland Interstadial 14 (GS-9 to GI-14, 43.8–54.2 ka BP). These time bins represent a trade-off between temporal resolution and sample size within each time period. The Subdivision of Late Pleistocene and Holocene time periods was based on current definitions of stratigraphic stages[92,93].

**Paleoecological model.** To analyze changes in the trophic position of the brown bear during the past 55,000 years, we used a Bayesian hierarchical model that consisted of two sub-models. Sub-model I was used to estimate the trophic position of the brown bear within each time period using the $\delta^{15}N$ values of red deer collagen as a dietary baseline for a strict herbivore. The model accounted for the effects of altitude[39], the type of sampled material (bones or teeth)[38] on $\delta^{15}N$ values of brown bears and red deer, and propagated all uncertainties associated with these factors as well as with trophic fractionation[38,40] through the entire model. Sub-model II then related the estimated trophic position to NPP and growing season length. Below, we describe the model in detail.

First, sub-model I estimated the average difference between the isotopic value of tooth and bone material based on paired samples of bone and teeth from the same individuals of different species of large carnivores ($n = 35$)[28,38]:

$$T_i \sim \text{Normal}\left(\mu_T, \sigma_T^2\right) \tag{14}$$

$$\sigma_T^2 \sim \text{scaled inverse} - \text{Gamma}(s, \text{df}) \tag{15}$$

$$\mu_T \sim \text{Normal}\left(0, 10^4\right) \tag{16}$$

We modeled the observed difference between the isotopic values of tooth and bone material $T_i$ using a normal distribution with mean $\mu_T$ and residual variance $\sigma_T^2$. We used an uninformative normal prior with a mean of 0 and a variance of $10^4$ for $\mu_T$ and a weakly informative scaled inverse gamma prior with scale $s = 1$ and two degrees of freedom (d.f. = 2) for $\sigma_T^2$.

Second, sub-model I estimated the effect of elevation on the isotopic values based on data that relates $\delta^{15}N$ (‰) of plants and herbivorous animals to elevation ($n = 69$)[28,39]:

$$\mu_{Ni} = \text{type}_{j[i]} + \beta_E E_i \tag{17}$$

$$N_i \sim \text{Normal}\left(\mu_{Ni}, \sigma_N^2\right) \tag{18}$$

$$\sigma_N^2 \sim \text{scaled inverse} - \text{Gamma}(s, \text{df}) \tag{19}$$

$$\text{type}_j \sim \text{Normal}\left(0, 10^4\right) \tag{20}$$

$$\beta_E \sim \text{Normal}\left(0, 10^4\right) \tag{21}$$

where $\mu_{Ni}$ is the expected $\delta^{15}N$ value of the $i$th observation, $\text{type}_{j[i]}$ is the type of material (vegetation, sheep wool, cattle hair, or goat hair), $\beta_E$ is the effect of elevation, and $E_i$ is the elevation at which sample $i$ has been collected. We modeled the observed $\delta^{15}N$ values $N_i$ using a normal distribution with mean $\mu_{Ni}$ and variance $\sigma_N^2$. We used uninformative normal priors with a mean of 0 and a variance of $10^4$ for the effects of material type and elevation and a weakly informative scaled inverse gamma prior with scale $s = 1$ and two degrees of freedom (d.f. = 2) for $\sigma_N^2$.

Third, sub-model I used the estimates of $\beta_E$ and $\mu_T$ to estimate the mean bias-corrected $\delta^{15}N$ values for red deer (baseline) and brown bear

(consumer) in each of the eight time periods (Supplementary Fig. 5):

$$\mu_{Bi} = baseline_{j[i]} + \beta_E E_i + \mu_T T_i \tag{22}$$

$$B_i \sim \text{Normal}\left(\mu_{Bi}, \sigma_B^2\right) \tag{23}$$

$$\mu_{Ci} = consumer_{j[i]} + \beta_E E_i + \mu_T T_i \tag{24}$$

$$C_i \sim \text{Normal}\left(\mu_{Ci}, \sigma_C^2\right) \tag{25}$$

where $\mu_{Bi}$ and $\mu_{Ci}$ are the expected $\delta^{15}N$ values of red deer and brown bear samples, $baseline_j$ and $consumer_j$ are the mean bias-corrected nitrogen signatures of red deer and brown bear in time period $j$, respectively, $E_i$ is the elevation at which sample $i$ has been collected, and $T_i$ indicates whether the sample is a tooth ($T_i = 1$). We modeled the observed $\delta^{15}N$ values of red deer $B_i$ and brown bear $C_i$ using normal distributions with means $\mu_{Bi}$ and $\mu_{Ci}$ and variances $\sigma_B^2$ and $\sigma_C^2$, respectively. We used uninformative normal priors with a mean of 0‰ and a variance of $10^4$‰ for the mean bias-corrected $\delta^{15}N$ values ($baseline_j$ and $consumer_j$) and weakly informative scaled inverse gamma priors with scale $s = 1$ and two degrees of freedom (d.f. = 2) for the variances ($\sigma_B^2$ and $\sigma_C^2$):

$$\sigma_B^2, \sigma_C^2 \sim \text{scaled inverse} - \text{Gamma}(s, \text{df}) \tag{26}$$

$$baseline_j, consumer_j \sim \text{Normal}\left(0, 10^4\right) \tag{27}$$

The trophic position of the brown bear in period $j$ $TP_j$ was then estimated as the difference between the mean bias-corrected $\delta^{15}N$ values of brown bear and red deer, divided by the trophic discrimination factor $\mu_\Delta$ plus an offset for the trophic level of the baseline ($\lambda = 2$)[94]:

$$TP_j = \left(consumer_j - baseline_j\right)/\mu_\Delta + \lambda \tag{28}$$

The trophic discrimination factor was estimated based on data for pairs of large carnivores and their prey (n = 10 pairs)[28,38]:

$$\Delta_i \sim \text{Normal}\left(\mu_\Delta, \sigma_\Delta^2\right) \tag{29}$$

$$\sigma_\Delta^2 \sim \text{scaled inverse} - \text{Gamma}(s, \text{df}) \tag{30}$$

$$\mu_\Delta \sim \text{Normal}\left(0, 10^4\right) \tag{31}$$

We modeled the observed trophic discrimination factor $\Delta_i$ using a normal distribution with mean $\mu_\Delta$ and residual variance $\sigma_\Delta^2$. We used an uninformative normal prior with a mean of 0 and a variance of $10^4$ for $\mu_\Delta$ and a weakly informative-scaled inverse gamma prior with scale $s = 1$ and two degrees of freedom (d.f. = 2) for $\sigma_\Delta^2$.

In a final step, sub-model II related the estimated trophic position $TP_j$ in each period $j$ to the NPP and growing season length in each period:

$$\mu_{Pj} = \sum_k \beta_k \mathbf{X}_{jk} \tag{32}$$

$$TP_i \sim \text{Normal}\left(\mu_{Pj}, \sigma_{TP}^2\right) \tag{33}$$

$$\sigma_P^2 \sim \text{scaled inverse} - \text{Gamma}(s, \text{df}) \tag{34}$$

$$\beta \sim \text{Normal}\left(0, 10^4\right) \tag{35}$$

where $\mu_j$ is the expected value of the $j$th observation, $\mathbf{X}$ is a model matrix including an intercept and the two explanatory variables (NPP [$\log_{10}$-transformed] and growing season length) and an associated vector of parameters $\beta$. We used an uninformative normal prior with a mean of 0 and a variance of $10^4$ for $\beta$ and a weakly informative-scaled inverse gamma prior with scale $s = 1$ and two degrees of freedom (d.f. = 2) for the residual variance.

## Model implementation and assessment of model fit

We used posterior predictive checks to assess the global fit of the models to the data[95]. As a measure of global model fit, we computed a posterior predictive $P$-value (PPP) that compares the residual sum of squares (RSS) based on the observed data to RSS generated from the posterior predictive distribution of the model. Values of PPP close to 0.5 indicate that the model fits the observed data, whereas values close to 0 or 1 indicate the opposite. We report the marginal variance, $r_m^2$, i.e., explained by the fixed factors, as well as the conditional variance, $r_c^2$, i.e., explained by the fixed and random factors combined[96]. Finally, we provide the median of all model estimates along with 50 and 90% Equal-Tailed Intervals (ETIs) as measures of uncertainty[97]. As a measure of support for the effects of the explanatory variables, we calculated the fraction of posterior samples with the same sign as the median ($p_d$). This metric quantifies the probability that the effect of a given explanatory variable differs from zero.

For both models we provide partial residual plots to illustrate the relationships of trophic position with NPP and growing season length and for visual inspection of model fit to single observations. To enhance the interpretation of the partial residual plots, we standardized the two predictor variables to zero mean and unit variance so that the effect of each predictor variable is shown while conditioning on the mean value of other predictor variable.

All analyses were conducted in $R$ (version 4.4.2)[83]. The Bayesian models were implemented in $JAGS$ (version 4.3) and run in $R$ through the $rjags$ package (version 4-16)[98]. We ran five parallel chains for the models. Each chain was run for 201,000 iterations with an adaptive burn-in phase of 1000 iterations and a thinning interval of 100 iterations, resulting in 2000 samples per chain, corresponding to 10,000 samples from the posterior distribution. Initial values were drawn randomly from the prior distributions using package $LaplacesDemon$ (version 16.1.6)[99]. The chains were checked for convergence, temporal autocorrelation and effective sample size using the $coda$ package (version 0.19–4.1)[100]. The residuals were checked for normality and variance homogeneity. We assessed potential collinearity among predictors using variance inflation factors[101].

## Ethics statement

The original specimens analyzed in this study are curated in the National Historical Museums (Stockholm, Sweden), the Lund University Historical Museum (Lund, Sweden), the Nationalmuseum Jamtli (Östersund, Sweden), the Archeological and Ethnographic Museum (Łódź, Poland), the Department of Environmental Archeology and Human Paleoecology, Nicolaus Copernicus University (Toruń, Poland), the Institute of Archeology, Nicolaus Copernicus University (Toruń, Poland), the Department of Paleoenvironmental Research, Adam Mickiewicz University (Poznań, Poland), the Department of Paleozoology at the University of Wrocław (Wrocław, Poland), the Museum of Archeology, Wrocław City Museum (Wrocław, Poland), the Institute of Systematics and Evolution of Animals, Polish Academy of Sciences (Kraków, Poland), the National Museum (Prague, Czech Republic), the Institute of Archeology of the Czech Academy of Sciences (Prague, Czech Republic), the Moravian Museum (Brno, Czech Republic), and the Museum of Bohemian Karst (Beroun, Czech Republic). Permission to sample the specimens for stable isotope analysis and radiocarbon dating was formally granted by each of the respective institutions, in accordance with their ethical guidelines and collection management policies. All sampling was carried out under these approved protocols and in collaboration with local scientists and curators to ensure the preservation and integrity of the collections. Curators of the above institutes were actively involved in evaluating and approving the study design and contributed to discussions regarding both the scope of the research and the interpretation of the results. Permits for collection and storage of samples were issued to the Institute of Nature Conservation of the Polish Academy of Sciences in Kraków by the General Directorate for Environmental Protection (Poland; permit number: DZP-WG.6401.08.4.2014.JRO,kka; date of issue: 04.06.2014).

## Reporting summary

Further information on research design is available in the Nature Portfolio Reporting Summary linked to this article.

## Data availability

The data generated in this study have been deposited in *figshare*[28] with the identifier (https://doi.org/10.6084/m9.figshare.25671048). The original specimens analyzed in this study are deposited in the National Historical Museums (Stockholm, Sweden), the Lund University Historical Museum (Lund, Sweden), the Nationalmuseum Jamtli (Östersund, Sweden), the Archeological and Ethnographic Museum (Łódź, Poland), the Department of Environmental Archeology and Human Paleoecology, Nicolaus Copernicus University (Toruń, Poland), the Institute of Archeology, Nicolaus Copernicus University (Toruń, Poland), the Department of Paleoenvironmental Research, Adam Mickiewicz University (Poznań, Poland), the Department of Paleozoology at the University of Wrocław (Wrocław, Poland), the Museum of Archeology, Wrocław City Museum (Wrocław, Poland), the Institute of Systematics and Evolution of Animals, Polish Academy of Sciences (Kraków, Poland), the National Museum (Prague, Czech Republic), the Institute of Archeology of the Czech Academy of Sciences (Prague, Czech Republic), the Moravian Museum (Brno, Czech Republic), and the Museum of Bohemian Karst (Beroun, Czech Republic). Detailed information on the deposition (institution and accession number) and provenance of each specimen (locality, country, and geographic coordinates) is provided in the raw data file available via the *figshare* link (data/paleoData/paleoDatabase_20250903.xlsx).

## Code availability

The computer code to reproduce the analyses and to create the figures is available in *figshare*[28] with the identifier (https://doi.org/10.6084/m9.figshare.25671048).

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

## Acknowledgements

We wholeheartedly thank Keith A. Hobson, who passed away on October 2, 2024. His deep connection with nature, his passion, and his dedication to ecology and nature conservation, as well as his inexhaustible enthusiasm were and will remain an inspiration to us. This study was funded by the Norway grants under the Polish-Norwegian Research Program administered by the National Research Center for Research and Development in Poland (project GLOBE No POL-NOR/198352/85/2013). J.A. was funded by the German Academic Exchange Service in the framework of a post doctorate fellowship grant (DAAD, No 91568794) and by the German Research Foundation (DFG, AL 2017/2-1). S.A.F. was funded by the Leibniz Association (Leibniz Competition P52/2017).

## Author contributions

J.A. and N.S. conceived the study. J.A. compiled data from the literature. J.A., A.S., A.M., E.I., L.D., R.K., G.L., D.M., and J.W. collected material and took samples from museum collections for stable isotope analyses. H.B., D.G.D. and K.A.H. conducted stable isotope analysis. J.A. developed the analytical tools, processed and analyzed the data and wrote the initial draft of the manuscript with input from N.S., H.B., K.A.H. and S.A.F.; J.A., H.B., K.A.H., D.G.D., A.S., J.S., A.Z., A.M., E.I., L.D., R.K., G.L., D.M., J.W., T.Z.-K., S.A.F., E.R. and N.S. reviewed and edited subsequent versions of the manuscript.

## Funding

## Competing interests

The authors declare no competing interests.

## Additional information

¹Senckenberg Biodiversity and Climate Research Centre (SBiK-F), Senckenberganlage 25, Frankfurt am Main, Germany. ²Institute of Nature Conservation, Polish Academy of Sciences, Mickiewicza 33, Kraków, Poland. ³Senckenberg Centre for Human Evolution and Palaeoenvironment (SHEP), University of Tübingen, Tübingen, Germany. ⁴Department of Geosciences (Biogeology), University of Tübingen, Tübingen, Germany. ⁵Environment and Climate Change Canada, Department of Biology, University of Western Ontario, London, Canada. ⁶Faculty of Environmental Sciences and Natural Resource Management, Norwegian University of Life Sciences, Ås, Norway. ⁷Department of Natural Sciences and Environmental Health, University of South-Eastern Norway, Bø, Norway. ⁸Institute for Wildlife Biology and Game Management, University for Natural Resources and Life Sciences, Vienna, Austria. ⁹Department of Paleozoology, Faculty of Biological Sciences, University of Wrocław, Wrocław, Poland. ¹⁰Department of Archaeology and Ancient History, Lund University, Lund, Sweden. ¹¹Department of Cultural History and Collections, National Historical Museum, Stockholm, Sweden. ¹²Department of Natural Sciences and Archaeometry, Institute of Archaeology of the Czech Academy of Sciences, Prague, Czech Republic. ¹³Institute of Systematics and Evolution of Animals, Polish Academy of Sciences, Kraków, Poland. ¹⁴Department of Environmental Archaeology and Human Paleoecology, Institute of Archaeology, Nicolaus Copernicus University of Toruń, Toruń, Poland. ¹⁵Department of Palaeontology, National Museum, Prague, Czech Republic. ¹⁶Tatra National Park, Zakopane, Poland. ¹⁷Institute of Geosciences, Goethe University Frankfurt, Frankfurt am Main, Germany. ¹⁸German Centre for integrative Biodiversity Research (iDiv) Halle-Jena-Leipzig, Leipzig, Germany. ¹⁹Institute of Biodiversity, Ecology and Evolution, Friedrich Schiller University Jena, Jena, Germany. ²⁰Estacion Biológica de Doñana Consejo Superior de Investigaciones Científicas (CSIC), Sevilla, Spain. ²¹Departamento de Ciencias Integradas, Facultad de Ciencias Experimentales, Centro de Estudios Avanzados en Física, Matemáticas y Computación, Universidad de Huelva, Huelva, Spain. ²²Deceased: Keith A. Hobson. ✉e-mail: joerg.albrecht@senckenberg.de

