## [Transparent Peer Review file · Nature Communications]

Dynamic omnivory shapes the functional role of large carnivores under global change

Corresponding Author: Dr Jörg Albrecht

Version 0:

Reviewer comments:

Reviewer #1

(Remarks to the Author)

The manuscript provides evidence that bears can flexibly alter their trophic position (degree of omnivory) across large spatial and temporal gradients of resource availability. A global diet analysis showed that 7 bear species increased their consumption of lower trophic level resources (i.e. increasing omnivory) as NPP and growing season increased. Next, subfossil bone and tooth samples of brown bears were used to show that trophic positions decreased from the Late Pleistocene to Holocene as NPP and growing season increased.

These findings are noteworthy examples of dynamic omnivory as a behavioral response to changes in resource availability in both space and time. I imagine they will be of widespread interest to ecologists and will motivate future work in the field. The conclusions and claims are well supported. I did not note any flaws in the data analysis, interpretation or conclusions. The methods are sound and meet expected standards. There is sufficient detail provided in the methods and the supplement to reproduce the study. I do have a few minor clarifying questions and comments I've listed below. Overall, this is an interesting study that provides novel information about spatial and temporal flexibility in omnivory.

Specific Comments

1) Could seasonal diet changes between animal and plant prey influence your results?

In the global dietary analysis, "samples covered the entire seasonal activity period of the focal species", meaning that any seasonal diet changes would still allow you to say what the contribution of animal prey was for that location (because all seasons were considered), correct?

What was the temporal resolution of your $\delta^{15}\text{N}$ data from each bone and teeth sample? Does the $\delta^{15}\text{N}$ extracted from collagen reflect the lifetime average trophic position of that individual bear? This might be worth mentioning in the text.

2) For the temporal analysis, what was the number of individual European brown bears included in each of the 8 time bins? I could have missed this somewhere.

3) Line 282-286: When comparing trophic positions to NPP and growing season across locations, the studies included in this analysis likely spanned a large date range. Can you clarify that data were pulled for NPP and monthly temperature from the same year in which the samples were collected for each study location, to account for interannual variation in environmental conditions?

4) Fig 1 shows how omnivory can change from no omnivory to omnivory and back as resource availability increases. That makes sense, and I see how this is meant to show a continuous gradient of decreasing trophic position (especially with the grey diagonal arrows). But visually, it highlighted for me the extreme end cases of no omnivory, and gave me the sense that 'dynamic omnivory' refers to a shift from no omnivory to omnivory (and vice versa). The Figure caption also read to me as if these are 3 discrete stages and didn't emphasize the continuous nature of the omnivory (intraguild predation) stage that is so well highlighted by your empirical findings (bears do not rely exclusively on plants or animals but increase plant consumption across the gradient, while still feeding as omnivores on both prey types).

I wondered if adding a second intraguild predation motif might help show that dynamic omnivory also includes shifts in the

degree of omnivory. These two motifs could show how the O-R interaction strengthens as the O-H interaction weakens as NPP increases. Visually, this might help the reader connect your conceptual figure with your empirical findings.

Finally, this figure is not referenced again in the text that I could find. It would be worth adding one sentence to the results or discussion stating how your empirical findings relate to the conceptual framework.

5) Fig. 2 and Fig. 3 are very effective and well done. I noticed some small boxes on both figures that are likely symbols that didn't convert properly?

(Remarks on code availability)

Reviewer #2

(Remarks to the Author)

The authors have assembled an impressive dataset and then followed a thoughtful modelling approach that allowed them to overcome unavoidable data limitations, to robustly estimate how the trophic position of bears in the Ursidae Family (as exemplar omnivores) changes across space with the length of the growing season and annual net primary productivity. They also considered the influence of competition from sympatric bear species. They then followed a similar approach to examine this relationship on a Paleoecological scale. As the authors note, omnivores play an important role in multitrophic communities and changes in their trophic position can have implications for the stability and functioning of an ecosystem. Therefore, understanding what influences their trophic position is a key ecological question and this study addresses it in a very convincing way.

All sections of the manuscript are very well written and easy to follow, clearly introducing the topic and explaining the methods that were used to answer the central questions of the project. This is also true for the script for the data preparation and analysis!

My main criticism is that, in their intro and discussion, the authors have placed a lot of emphasis on the effects that resource availability may have on bear diet, to the near exclusion of other potential mechanisms. I would argue that one of them, direct effects of temperature on bears, is already embedded in the data they are analyzing. I would expect that annual net primary productivity is strongly dependent on growth season length. (A plot showing the relationship of the two would be a useful addition to the supplement – I know the authors checked for variance inflation; this is just to help readers understand all the relevant relationships). The total effect of GSL on diet would be largely mediated by NPP. So, with both of these variables as predictors, the model is estimating the “direct” effect of GSL on diet (i.e. unmediated by NPP) and the effect of NPP on diet, accounting for the common influence of GSL on both NPP and TP. Here is a causal diagram showing the main effects of their model:

NPP <- GSL -> TP

NPP -> TP

(I know the authors know all that; I am merely laying out so that we are all on the same page.)

I agree that NPP and partly GSL are measures of plant diet availability during the year. So the authors' conclusion, that bears rely less on animal diet because plant diet becomes increasingly available, is partly supported by these results.

However, GSL is also strongly related to mean annual temperature. So I would expect that part of the effect of GSL on bear diet that is unmediated by NPP, is really the direct effect of temperature on bear physiology and activity. I would suggest that the larger the number of months with below-zero temperatures, the more bears would need to rely on animal diet to build fat reserves. Then additionally, with higher temperatures, heat loss is no longer a problem; in fact, the reduction of heat dissipation might restrict activity, further promoting a shift from hunting to foraging. Looking at Figures 3a and 3c I think this is also true for the Paleo analysis; trophic position seems to track changes in GSL more than NPP. Here is a more complete diagram:

NPP <- GSL -> (PDA) -> TP

NPP -> (PDA) -> TP

GSL -> TP

where (PDA) is the (unobserved) plant diet availability. In the absence of PDA, this causal model simplifies to the previous one, with the caveat that the arrow from GSL to TP (and the corresponding model estimate) captures both the effects of GSL that relate to plant diet availability during the year, as well as direct effects of temperature on bear trophic activity. I do not think that these two are separable in practice, so I am not asking that the authors fit a different model. I believe their model is the best model one could fit. I am merely asking, assuming they agree with my interpretation, that they elaborate a bit on the more complex nature of that GSL->TP effect.

I know this complicates the take-home message of this work, but it is an important aspect of how omnivory relates to ecosystem processes; one of these mechanisms is a within-system adaptation (trophic behaviour influenced by resource availability), while the other one is external forcing (trophic behaviour influenced by temperature).

I understand the logic behind the exclusion of *U. maritimus* but it would be interesting to have its trophic position included in the figures as a reference point, as the most carnivorous species in the Family; from a brief look at the diet data, they do consume some plant material.

Figure 2c,d: I cannot reconcile the information in lines 354-358 of Methods about the compression to (0.00034-0.99934), with the scatterplot of these two panels. The trophic position data seem to be within a (0.1-0.8) range. Also, in panel B, the range for values is larger. Is it because this panel represents not compressed data?

L114-115: I might have missed it, but I did not find how this overlap was estimated and where it was used. It seems only mentioned in the SI with the dominant and subordinate factors. Also, would it not make sense to make this a continuous

variable reflecting the proportion of overlap?

L120-121: could you add a measure of representing an (any kind of) effect size to the posterior probabilities?

Supplement: Could you add a plot showing the relationship of NPP to GSL?

(Remarks on code availability)

Code is executable and extremely well commented. We did not carefully check for potential mistakes in it though

Reviewer #3

(Remarks to the Author)

(Remarks on code availability)

Reviewer #4

(Remarks to the Author)

The study presents a novel and innovative application of the dynamic omnivory hypothesis (Figure 1) to large, terrestrial, mammalian carnivores by investigating spatial and temporal dietary variation in ursids. Overall the study was well organized and composed, and the methods and results were clearly explained. The references were comprehensive, and this paper has potential to serve as solid resource for future work on this topic. My substantive comments and questions are in relation to the interpretation of the findings – see below.

For the modern spatial analysis of global bear species TP I had three comments and questions;

- 1) It appears in Figure 2 that really only brown bears are “dynamically adapting” their species-specific TP to resource availability... for 5 of the other species besides the black bear, there does not look like there is substantial TP variation to explain. And while the black bear TP distribution does extend, it is only due to a few individuals. To what extent does the spatial analysis and interpretation apply to bear species other than brown (and possibly black) bears?
- 2) Along the same line of inquiry, the term dynamic implies an ability to respond – how does bear species morphology and body size factor into the interpretations? Are all 7 species able to respond to resource changes by altering TP, or are some more constrained than others by evolutionary legacy? This could be an important consideration before the model results are applied to all bear species.
- 3) The model accounts for sympatric competition among bear species, but I did not see any discussion of the potential confounding effects of interactions with other members of the carnivore and omnivore guilds? Could variation in community assemblage influence the response of brown bears to changing resources?

For the fossil temporal analysis of European brown bear TP I had two comments and questions;

- 1) Dietary resources of marine origin, such as anadromous fish or coastal invertebrates, could influence the interpretation of TP from brown bear d15N values (relative to red deer baseline, Figure 3). Is this something the authors considered? Specifically, are there spatial patterns in fossil location (proximity to coastline, river systems with geohistorical anadromous fish presence, etc.) that interact with the temporal bear TP patterns? The trend in brown bear TP through time is convincingly correlated to climate and NPP, but given the importance of marine resources for populations of brown bears today, it could be worth addressing in the discussion of these results.
- 2) The covariation in d15N and d13C values of marine diet resources (both are generally higher than terrestrial) could be another way to query the isotopic data to identify individual bears for which TP (estimated from d15N values) might be influenced by marine resources, not solely climate/NPP factors. The methods refer to d13C analyses, but I did not see these data presented, is this something the authors considered?

(Remarks on code availability)

Version 1:

Reviewer comments:

Reviewer #1

(Remarks to the Author)

(Remarks on code availability)

Reviewer #2

(Remarks to the Author)

The revised manuscript addresses all our previous concerns and we look forward to seeing it published.

(Remarks on code availability)

Reviewer #3

(Remarks to the Author)

(Remarks on code availability)

Reviewer #4

(Remarks to the Author)

All of my comments and questions have been thoughtfully and thoroughly addressed in the revised version of the manuscript.

(Remarks on code availability)

Responses to reviewers

We would like to thank the Reviewers for their constructive comments. Below we provide a point-by-point response to the comments, which are highlighted in blue font. To facilitate the review process, we have highlighted the changes made to the manuscript in red font in the Word document. Please note that line numbers in our responses refer to lines in the revised version of the manuscript. At the end of our responses, we provide a list of the cited literature.

Reviewer comments

Reviewer #1 (Remarks to the Author):

The manuscript provides evidence that bears can flexibly alter their trophic position (degree of omnivory) across large spatial and temporal gradients of resource availability. A global diet analysis showed that 7 bear species increased their consumption of lower trophic level resources (i.e. increasing omnivory) as NPP and growing season increased. Next, subfossil bone and tooth samples of brown bears were used to show that trophic positions decreased from the Late Pleistocene to Holocene as NPP and growing season increased.

These findings are noteworthy examples of dynamic omnivory as a behavioral response to changes in resource availability in both space and time. I imagine they will be of widespread interest to ecologists and will motivate future work in the field. The conclusions and claims are well supported. I did not note any flaws in the data analysis, interpretation or conclusions. The methods are sound and meet expected standards. There is sufficient detail provided in the methods and the supplement to reproduce the study. I do have a few minor clarifying questions and comments I've listed below. Overall, this is an interesting study that provides novel information about spatial and temporal flexibility in omnivory.

Response 1.1: We are grateful for the reviewer's encouraging remarks and appreciation of our work.

Specific Comments

1) Could seasonal diet changes between animal and plant prey influence your results?

In the global dietary analysis, "samples covered the entire seasonal activity period of the focal species", meaning that any seasonal diet changes would still allow you to say what the

contribution of animal prey was for that location (because all seasons were considered), correct?

Response 1.2: It is true that the diet of the species also varies seasonally. However, these differences do not influence our results, as we only included studies that covered the entire seasonal activity period of the focal species. We used the average annual diet, because it is more representative of the diet than seasonal snapshots and only this approach allows for meaningful comparisons across study locations. Previous work has shown that this approach integrates very well across seasonal dietary changes (e.g., in temperate or boreal biomes)¹. In fact, using seasonal dietary snapshots instead of average annual diet could lead to detection of spurious differences, for example, if the spring diet in one location is compared with autumn diet in another location. We added a sentence into the main text outlining our rationale for using annual diets (**see lines 104-106**). The sentence reads: “...*We used annual diets, which integrate foraging behavior across the entire active season of the focal species, to enable meaningful comparisons across study locations. ...*”.

What was the temporal resolution of your $\delta^{15}\text{N}$ data from each bone and teeth sample? Does the $\delta^{15}\text{N}$ extracted from collagen reflect the lifetime average trophic position of that individual bear? This might be worth mentioning in the text.

Response 1.3: In long-lived large mammals, adult bone collagen reflects the average isotopic composition of the diet over several years preceding death, whereas collagen in tooth dentine primarily accumulates during the period of tooth development². Since brown bears possess brachydont teeth dental growth ceases early in life, during a stage when a substantial proportion of nutrients is still derived from lactation². Consequently, collagen extracted from dentine may exhibit higher $\delta^{15}\text{N}$ values than bone collagen from the same individual². We accounted for this effect in the analyses by correcting the isotope ratio of teeth using data from the literature² on the difference in $\delta^{15}\text{N}$ values between collagen from tooth and bone material of the same individuals. Following your suggestion, we added this explanation to the methods section of the manuscript (**see lines 463-471**).

2) For the temporal analysis, what was the number of individual European brown bears included in each of the 8 time bins? I could have missed this somewhere.

Response 1.4: The sample sizes for the European brown bear and red deer (the isotopic baseline for a strict herbivore) are indicated at the bottom in Fig. 3c. The information is also highlighted in the figure legend. To make the link of the sample sizes to the species clearer, we added the scientific names to the legend (see lines 932-933). It now reads: “...*Sample sizes for brown bear ($n_U. arctos$) and red deer ($n_C. elaphus$) in each time bin are given at the bottom of the panel....*”.

3) Line 282-286: When comparing trophic positions to NPP and growing season across locations, the studies included in this analysis likely spanned a large date range. Can you clarify that data were pulled for NPP and monthly temperature from the same year in which the samples were collected for each study location, to account for interannual variation in environmental conditions?

Response 1.5: Unfortunately, we were unable to obtain annually resolved data on NPP and growing season length for the macroecological analysis. As a result, our model does not account for interannual variation in these environmental factors. Instead, we used NPP estimates from Terra MODIS17A3 and calculated growing season length based on CHELSA climatologies of near-surface air temperature. Both datasets offer excellent spatial resolution and integrate data over extended time periods, minimizing the influence of individual extreme years on our analysis and conclusions (temporal coverage: CHELSA – 30-year average; Terra MODIS17A3 – 16-year average).

Nevertheless, given the large spatial scale of our macroecological analysis, we expect that geographic differences in climate between populations are more pronounced and have a greater influence on observed geographic patterns in trophic position than interannual climatic variability. Furthermore, we expect the effects of NPP and growing season length on trophic position in our macroecological analysis to be conservative estimates. If annually resolved predictor variables were available, these effects might be even stronger, as they would capture interannual climatic variation in addition to geographic variation in climatic conditions.

4) Fig 1 shows how omnivory can change from no omnivory to omnivory and back as resource availability increases. That makes sense, and I see how this is meant to show a continuous gradient of decreasing trophic position (especially with the grey diagonal arrows). But visually, it highlighted for me the extreme end cases of no omnivory, and gave me the sense that ‘dynamic omnivory’ refers to a shift from no omnivory to omnivory (and vice versa). The Figure caption also read to me as if these are 3 discrete stages and didn’t emphasize the continuous nature of the omnivory (intraguild predation) stage that is so well highlighted by your empirical findings (bears do not rely exclusively on plants or animals but increase plant consumption across the gradient, while still feeding as omnivores on both prey types).

I wondered if adding a second intraguild predation motif might help show that dynamic omnivory also includes shifts in the degree of omnivory. These two motifs could show how the O-R interaction strengthens as the O-H interaction weakens as NPP increases. Visually, this might help the reader connect your conceptual figure with your empirical findings.

Finally, this figure is not referenced again in the text that I could find. It would be worth adding one sentence to the results or discussion stating how your empirical findings relate to the conceptual framework.

Response 1.6: We followed your suggestion and modified the figure by adding a second intraguild predation motif to highlight the continuous nature of dynamic omnivory. We also added a sentence to the figure caption to highlight the shift in relative interaction strengths across the environmental gradients on the x-axis (see revised Fig. 1). Moreover, we added references to this figure in the results section to relate our empirical findings back to the conceptual framework (see lines 172-174). The section reads: “...*These results are in strong agreement with the ‘dynamic omnivory’ hypothesis, predicting that omnivores adapt their trophic position to resource availability and growing season length (Fig. 1).*...”

5) Fig. 2 and Fig. 3 are very effective and well done. I noticed some small boxes on both figures that are likely symbols that didn’t convert properly?

Response 1.7: Thank you for pointing to problems with figure conversion. Upon uploading the revised version of our manuscript, we checked that symbols converted properly.

Reviewer #2 (Remarks to the Author):

The authors have assembled an impressive dataset and then followed a thoughtful modelling approach that allowed them to overcome unavoidable data limitations, to robustly estimate how the trophic position of bears in the Ursidae Family (as exemplar omnivores) changes across space with the length of the growing season and annual net primary productivity. They also considered the influence of competition from sympatric bear species. They then followed a similar approach to examine this relationship on a Paleoecological scale. As the authors note, omnivores play an important role in multitrophic communities and changes in their trophic position can have implications for the stability and functioning of an ecosystem. Therefore, understanding what influences their trophic position is a key ecological question and this study addresses it in a very convincing way.

All sections of the manuscript are very well written and easy to follow, clearly introducing the topic and explaining the methods that were used to answer the central questions of the project. This is also true for the script for the data preparation and analysis!

Response 2.1: We appreciate the reviewer's positive assessment of our study.

My main criticism is that, in their intro and discussion, the authors have placed a lot of emphasis on the effects that resource availability may have on bear diet, to the near exclusion of other potential mechanisms. I would argue that one of them, direct effects of temperature on bears, is already embedded in the data they are analyzing. I would expect that annual net primary productivity is strongly dependent on growth season length. (A plot showing the relationship of the two would be a useful addition to the supplement – I know the authors checked for variance inflation; this is just to help readers understand all the relevant relationships). The total effect of GSL on diet would be largely mediated by NPP. So, with both of these variables as predictors, the model is estimating the “direct” effect of GSL on diet (i.e. unmediated by NPP) and the effect of NPP on diet, accounting for the common influence of GSL on both NPP and TP. Here is a causal diagram showing the main effects of their model:
NPP <- GSL -> TP

NPP -> TP

(I know the authors know all that; I am merely laying out so that we are all on the same page.) I agree that NPP and partly GSL are measures of plant diet availability during the year. So the

authors' conclusion, that bears rely less on animal diet because plant diet becomes increasingly available, is partly supported by these results. However, GSL is also strongly related to mean annual temperature. So I would expect that part of the effect of GSL on bear diet that is unmediated by NPP, is really the direct effect of temperature on bear physiology and activity. I would suggest that the larger the number of months with below-zero temperatures, the more bears would need to rely on animal diet to build fat reserves. Then additionally, with higher temperatures, heat loss is no longer a problem; in fact, the reduction of heat dissipation might restrict activity, further promoting a shift from hunting to foraging. Looking at Figures 3a and 3c I think this is also true for the Paleo analysis; trophic position seems to track changes in GSL more than NPP. Here is a more complete diagram:

NPP <- GSL -> (PDA) -> TP

NPP -> (PDA) -> TP

GSL -> TP

where (PDA) is the (unobserved) plant diet availability. In the absence of PDA, this causal model simplifies to the previous one, with the caveat that the arrow from GSL to TP (and the corresponding model estimate) captures both the effects of GSL that relate to plant diet availability during the year, as well as direct effects of temperature on bear trophic activity. I do not think that these two are separable in practice, so I am not asking that the authors fit a different model. I believe their model is the best model one could fit. I am merely asking, assuming they agree with my interpretation, that they elaborate a bit on the more complex nature of that GSL->TP effect.

I know this complicates the take-home message of this work, but it is an important aspect of how omnivory relates to ecosystem processes; one of these mechanisms is a within-system adaptation (trophic behaviour influenced by resource availability), while the other one is external forcing (trophic behaviour influenced by temperature).

Response 2.2: Thank you for this constructive and critical comment. We added the point related to physiological effects of temperature on foraging behaviour to the introduction (see lines 74-76). The section in the introduction now reads: "... Moreover, prolonged growing seasons due to climate warming³ can reduce seasonal bottlenecks in NPP and may release consumers from energetic constraints by reducing the metabolic costs of maintenance and foraging in warmer environments^{4,5}...". In the revised version of the manuscript, we now also explicitly highlight in the discussion section (see lines 179-186 and 229) physiological

temperature effects as a potential mechanism underlying altered foraging behavior. The section reads: “...*We attribute the observed trophic adaptation to three nonexclusive mechanisms: (i) tracking seasonal peaks in plant productivity (e.g., fruit and seed production), facilitated by the mobility of these large omnivores^{6,7}; (ii) shifts in foraging preferences if foraging for plant resources is energetically more efficient than hunting or scavenging, due to lower search and handling times in dense vegetation^{8,9}; and (iii) preferential foraging for plant resources in warmer environments, where reduced metabolic costs of maintenance and foraging alleviate energetic constraints^{4,5}....*”

I understand the logic behind the exclusion of *U. maritimus* but it would be interesting to have its trophic position included in the figures as a reference point, as the most carnivorous species in the Family; from a brief look at the diet data, they do consume some plant material.

Response 2.3: Thank you for this comment. We also initially considered adding the polar bear to the analysis as a reference point. However, this brings up several problems noted in the manuscript: First, the scat samples collected during summer at coasts are not representative of the species' diet, because polar bears are effectively fasting during this period and acquire energy mainly from predation of marine mammals on sea ice during autumn, winter, and spring^{10–12}. More recent work indicates that observed population declines of polar bears are in fact associated with food shortage during prolonged periods ashore, suggesting that they cannot persist solely on plant-based diets¹³. Moreover, those dietary studies describing winter diets of polar bears, have been conducted using fatty acid analysis of adipose tissue¹⁴. Therefore, these results are not directly comparable to the micro histological analyses of scat and stomach contents used in our study. Finally, we do not have comparable data on net primary productivity or a comparable variable describing growing season length for this species, as polar bears spend most of the year on sea ice and only fast on land during summer. Even though we appreciate the suggestion of the reviewer, we would like to avoid including the polar bear as a reference point into the analysis for the reasons stated above.

Figure 2c,d: I cannot reconcile the information in lines 354-358 of Methods about the compression to (0.00034-0.99934), with the scatterplot of these two panels. The trophic position data seem to be within a (0.1-0.8) range. Also, in panel B, the range for values is larger. Is it because this panel represents not compressed data?

Response 2.4: The panels c,d show component + residual plots, which are conceptually similar to partial residual plots. Therefore, these two panels do not show the raw data, but the marginal relationships between the response variable and each predictor variable after accounting for the effects of other variables in the model. This is why the distribution of the data in these plots is not the same as in panel b, in which the raw data is shown. We already highlighted this in the captions of Figure 2 and 3 (see lines 908-911 and 934-935). We hope that this clarifies the question of the reviewer.

L114-115: I might have missed it, but I did not find how this overlap was estimated and where it was used. It seems only mentioned in the SI with the dominant and subordinate factors. Also, would it not make sense to make this a continuous variable reflecting the proportion of overlap?

Response 2.5: Thank you for this comment. We added an explanation of how we quantified co-occurrence to the methods section (see lines 327-332). The section reads: “...*We quantified species co-occurrences using IUCN species distribution maps*¹⁵. For the geographic records of each species, we determined whether a location fell within the geographic range of one or more other bear species. Then we created a categorical variable with three levels (no co-occurrence, subordinate, and dominant), indicating whether the focal species at a given location co-occurred with another bear species and, if so, whether it was smaller (subordinate) or larger (dominant) than the co-occurring species....”

While we appreciate the advantages of using a quantitative measure of proportional range overlap in the context of species-centered analyses, this was not meaningful in the context of our population-centered analysis, because each population could have different values for the co-occurrence variable, so that some populations could be located within the geographic range of another species while this was not the case for other populations (see Table 1 below).

Therefore, we used a categorical variable indicating the co-occurrence and the potential type of interaction with other bear species based on their body mass.

Table 1. Summary of co-occurrence pattern across the populations of each bear species included in the study.

Species	Co-occurrence		
	no	Focal species smaller (subordinate)	Focal species larger (dominant)
Ursus arctos	71	6	18
Ursus americanus	52	11	0
Ursus thibetanus	24	4	0
Helarctos malayanus	2	0	0
Melursus ursinus	11	0	3
Tremarctos ornatus	6	0	0
Ailuropoda melanoleuca	0	0	2

L120-121: could you add a measure of representing an (any kind of) effect size to the posterior probabilities?

Response 2.6: Following your suggestion, we added effect size measures to the posterior probabilities. In particular, we added the median and the 90% Equal-Tailed Intervals (ETI) in line with the information given in the supplementary materials (see lines 126-128, 140-142, and 165-166).

Supplement: Could you add a plot showing the relationship of NPP to GSL?

Response 2.7: We added plots of these two variables based on the macroecological and paleoecological data to the supplement and an explanatory sentence to the respective parts of the methods section (see new Supplementary Figs. 1 and 4; lines 322-325 and 490-493).

Reviewer #2 (Remarks on code availability):

Code is executable and extremely well commented. We did not carefully check for potential mistakes in it though

Response 2.8: Thank you for the positive feedback on the code.

Reviewer #3 (Remarks to the Author):

Response 3.1: We appreciate your time and effort in co-reviewing our manuscript.

Reviewer #4 (Remarks to the Author):

The study presents a novel and innovative application of the dynamic omnivory hypothesis (Figure 1) to large, terrestrial, mammalian carnivores by investigating spatial and temporal dietary variation in ursids. Overall the study was well organized and composed, and the methods and results were clearly explained. The references were comprehensive, and this paper has potential to serve as solid resource for future work on this topic. My substantive comments and questions are in relation to the interpretation of the findings – see below.

Response 4.1: Thank you for your valuable time and thoughtful review of our manuscript.

For the modern spatial analysis of global bear species TP I had three comments and questions;

1) It appears in Figure 2 that really only brown bears are “dynamically adapting” their species-specific TP to resource availability... for 5 of the other species besides the black bear, there does not look like there is substantial TP variation to explain. And while the black bear TP distribution does extend, it is only due to a few individuals. To what extent does the spatial analysis and interpretation apply to bear species other than brown (and possibly black) bears?

Response 4.2: Thank you for this comment. With the macroecological analysis we aimed to test (i) whether differences in trophic position between bear species converge on the same general relationship with NPP and growing season length, and (ii) whether geographic differences in trophic position between populations of the same species can be attributed to geographic differences in NPP and growing season length between these populations.

With respect to question (i) our analysis nicely illustrates that there is a general relationship of the trophic position of bears with NPP and growing season length (Fig. 2). This indicates that the realized trophic positions of the different bear species are largely a result of the environmental conditions encountered by each species.

With respect to question (ii) our analysis provides the strongest evidence for those species with broad distributions that consequently also experience a broad range of environmental conditions to which they need to adapt (i.e., brown bears, American black bears, Asiatic black bears, and sloth bears; see **Fig. 2a,b in main text**). However, our conclusions regarding the adaptive capacity of those species with smaller geographic distributions are weaker (i.e., sun bear, spectacled bear, and giant panda). Yet, previous work has shown that the majority of extant bear species exhibit significant dietary flexibility, enabling them to subsist on plant- or animal-based diets for prolonged periods of time depending on resource availability^{16–19}. Brown bears, American black bears, and Asiatic black bears are known to consume a wide range of plant and animal matter¹⁸. Likewise, sloth bears and sun bears have been reported to adjust their diets seasonally between insects and fruits^{17,20,21}. And even spectacled bears that strongly rely on plant-based diets have been reported to scavenge or predate on smaller mammals¹⁷. This is also reflected in relatively weak morphological skull adaptations of these species, which have been interpreted as resulting from selective pressures favoring dietary flexibility over specialization^{17–19,22}.

Polar bears and giant pandas are indeed exceptions in this context as they represent the extremes of the carnivory–omnivory–herbivory gradient in our study: Polar bears are almost strictly carnivorous, preying on a range of marine mammals on sea ice¹⁴, and consume vegetative material only in southern areas where they are forced ashore during summer by melted sea ice^{10,16,23}. Observed population declines associated with food shortage during prolonged periods ashore suggest that they cannot persist solely on terrestrial diets¹³. At the other extreme, the diet of giant pandas is limited almost entirely to bamboo^{24,25}, and the species only occasionally scavenges and predate on small mammals¹⁷. Therefore, we expect that only those species that are omnivorous and have been shown to switch seasonally between animal- and plant-based diets depending on their availability are able to adapt their trophic position to changes in resource availability. To clarify this aspect, we added this notion to the discussion section of the manuscript (see **lines 186–198**). The section reads: “...*These results are supported by previous work reporting trophic adaptations to seasonal and annual variability in resource availability within populations of six of the seven terrestrial bear species*^{22,17,19,9,26,16,18}. *The only exceptions among ursids in this context are the polar bear and the giant panda (*Ailuropoda melanoleuca*), which represent the extremes of the carnivory–omnivory–herbivory gradient in our study (Fig. 1): Polar bears are almost strictly carnivorous, preying on a range of marine mammals on sea ice¹⁴ and consuming plant resources (e.g.,*

berries) only in southern areas where they are forced ashore during the ice-free period in summer^{10,16,23}. At the other extreme, giant pandas are strictly herbivorous, consuming almost exclusively bamboo^{24,25} and only occasionally scavenging or preying on small mammals¹⁷. Therefore, we expect that only those species that are omnivorous and have been shown to switch seasonally between animal- and plant-based diets depending on availability are able to adapt their trophic position to changes in resource availability. ...”. We hope that this addition to the discussion addresses the concerns of the reviewer.

2) Along the same line of inquiry, the term dynamic implies an ability to respond – how does bear species morphology and body size factor into the interpretations? Are all 7 species able to respond to resource changes by altering TP, or are some more constrained than others by evolutionary legacy? This could be an important consideration before the model results are applied to all bear species.

Response 4.3: As mentioned above, except for the polar bear and the giant panda, all other extant bear species have maintained a remarkable degree of dietary flexibility, which allows them to subsist on plant or animal diets depending on resource availability^{16–19}. Importantly, the giant panda is the only ursid species that shows strong adaptations in skull morphology to its specialized bamboo diet^{17,19}. For all other ursids, including polar bears, skull morphology is less derived than would be expected by their diets^{17,19}, suggesting that selective pressures on skull morphology are relatively weak¹⁷. Instead, it has been suggested in the literature that the relatively weak adaptations for carnivory and herbivory observed in bears may be the result of selection for the ability to cope with temporal fluctuations in dietary components¹⁹. We elaborated on the potential constraints imposed by morphology in the discussion section (**see lines 234-249**). The section reads: “... *Despite the signal of trophic adaptation of populations to environmental change in our study, it remains unclear to what extent species that show morphological adaptations to specific diet types (e.g., herbivory in the giant panda)^{17,19,22,27} or populations that occur at the physiological or ecological limits of their species’ range in extreme environments (e.g., arctic, desert, or alpine environments) are able to cope with future changes in environmental conditions. The adaptive capacity of these species and populations will depend on the magnitude and velocity of environmental change, as well as on whether alternative food sources meet their nutritional requirements^{6,28}. For instance, it has been suggested that the switch of polar bears from marine prey to terrestrial food sources (e.g.,*

berries or bird nests) in response to reductions in sea-ice extent is associated with declines in body size, body condition, reproduction, and population sizes¹³. Similarly, due to their specialization on bamboo, the capacity of giant pandas to adjust their trophic niche to variations in resource availability is likely limited²⁴. Investigating how the interplay of behavioral, morphological, and physiological factors constrains or facilitates the trophic flexibility of omnivores may provide further insights into their adaptive capacity to environmental change. ...”. We hope that this addresses the concerns of the reviewer.

3) The model accounts for sympatric competition among bear species, but I did not see any discussion of the potential confounding effects of interactions with other members of the carnivore and omnivore guilds? Could variation in community assemblage influence the response of brown bears to changing resources?

Response 4.4: We did not account for co-occurrence with other large carnivores in the analysis (e.g., felids or canids), as bears have evolved from carnivores with high-protein diets to omnivores that consume low-protein diets, thus, reducing interspecific competition with other large carnivores²⁹. Moreover, bears are well-documented kleptoparasites, often monopolizing carrion resources and significantly limiting carrion access and predatory behaviour in other large carnivores (e.g., wolves, cougars, and pumas)^{30–36}. In interspecific encounters with other carnivores, bears typically dominate due to their large body size or are able to defend themselves by charging, as observed in encounters with tigers (*Panthera tigris*)^{37–39}. In extreme cases of interspecific killing, bears are typically the killer species, while rare instances of bears being killed usually involve denning adults, cubs of the year, or aggressive encounters with tigers in Indian tiger reserves and the Russian Far East^{37–42}. We added this justification to the methods section of the manuscript (see lines 333-343).

For the fossil temporal analysis of European brown bear TP I had two comments and questions;

1) Dietary resources of marine origin, such as anadromous fish or coastal invertebrates, could influence the interpretation of TP from brown bear d15N values (relative to red deer baseline, Figure 3). Is this something the authors considered? Specifically, are there spatial patterns in fossil location (proximity to coastline, river systems with geohistorical anadromous fish presence, etc.) that interact with the temporal bear TP patterns? The trend in brown bear TP

through time is convincingly correlated to climate and NPP, but given the importance of marine resources for populations of brown bears today, it could be worth addressing in the discussion of these results.

2) The covariation in d15N and d13C values of marine diet resources (both are generally higher than terrestrial) could be another way to query the isotopic data to identify individual bears for which TP (estimated from d15N values) might be influenced by marine resources, not solely climate/NPP factors. The methods refer to d13C analyses, but I did not see these data presented, is this something the authors considered?

Response 4.5: Thank you for these important comments. As suggested, we used the d15N and d13C stable isotope ratios to rule out any influence of marine diet resources, such as anadromous fish or coastal invertebrates. To do so, we compared the stable isotope ratios of the brown bear in our data to those of other a range of different marine mammal species from the literature⁴³. Importantly, we found no indication of consumption of marine resources for the brown bear samples in our data, as the d15N and d13C values are much lower than would be expected, if the respective brown bear individuals had consumed marine resources (see new **Supplementary Fig. 3**). We added this analysis to the methods section of the main text and the supplementary materials (see new **Supplementary Fig. 3**), and briefly refer to the analysis in the results section (see lines **150-152 and 473-481**). The section in the methods reads: “...*To rule out the consumption of marine resources (e.g., anadromous fish) by brown bears, we compared the $\delta^{13}\text{C}$ and $\delta^{15}\text{N}$ values of the brown bear material with those of a range of marine mammal species from the literature⁴³ (Supplementary Fig. 3). The $\delta^{13}\text{C}$ and $\delta^{15}\text{N}$ values of brown bear material were much lower than those of material from marine mammals, but very similar to those of the red deer material (Supplementary Fig. 3), indicating negligible consumption of marine resources by brown bears in our data. Therefore, we estimated the trophic position of brown bears based on $\delta^{15}\text{N}$ of collagen through time using the $\delta^{15}\text{N}$ values of red deer from the same region as a dietary baseline for a strict herbivore...*”.

References

1. Vulla, E. *et al.* Carnivory is positively correlated with latitude among omnivorous mammals: evidence from brown bears, badgers and pine martens. *Ann. Zool. Fenn.* **46**, 395–415 (2009).

2. Bocherens, H. Isotopic tracking of large carnivore palaeoecology in the mammoth steppe. *Quat. Sci. Rev.* **117**, 42–71 (2015).
3. Liu, H., Lu, C., Wang, S., Ren, F. & Wang, H. Climate warming extends growing season but not reproductive phase of terrestrial plants. *Glob. Ecol. Biogeogr.* **30**, 950–960 (2021).
4. Anderson, K. J. & Jetz, W. The broad-scale ecology of energy expenditure of endotherms: Constraints on endotherm energetics. *Ecol. Lett.* **8**, 310–318 (2005).
5. Arim, M., Bozinovic, F. & A. Marquet, P. On the relationship between trophic position, body mass and temperature: reformulating the energy limitation hypothesis. *Oikos* **116**, 1524–1530 (2007).
6. Bartley, T. J. *et al.* Food web rewiring in a changing world. *Nat. Ecol. Evol.* **3**, 345–354 (2019).
7. Gutgesell, M. K. *et al.* On the dynamic nature of omnivory in a changing world. *BioScience* **72**, 416–430 (2022).
8. Welch, C. A., Keay, J., Kendall, K. C. & Robbins, C. T. Constraints on frugivory by bears. *Ecology* **78**, 1105–1119 (1997).
9. Deacy, W. W. *et al.* Phenological synchronization disrupts trophic interactions between Kodiak brown bears and salmon. *Proc. Natl. Acad. Sci.* **114**, 10432–10437 (2017).
10. Hobson, K. A. & Stirling, I. Low variation in blood $\delta^{13}\text{C}$ among Hudson bay polar bears: Implications for metabolism and tracing terrestrial foraging. *Mar. Mammal Sci.* **13**, 359–367 (1997).
11. Ramsay, M. A. & Hobson, K. A. Polar Bears Make Little Use of Terrestrial Food Webs: Evidence from Stable-Carbon Isotope Analysis. (2008).
12. Hobson, K. A., Stirling, I. & Andriashek, D. S. Isotopic homogeneity of breath CO₂ from fasting and berry-eating polar bears: implications for tracing reliance on terrestrial foods in a changing Arctic. *Can. J. Zool.* **87**, 50–55 (2009).
13. Archer, L. C., Atkinson, S. N., Lunn, N. J., Penk, S. R. & Molnár, P. K. Energetic constraints drive the decline of a sentinel polar bear population. *Science* **387**, 516–521 (2025).
14. Thiemann, G. W., Iverson, S. J. & Stirling, I. Polar bear diets and Arctic marine food webs: insights from fatty acid analysis. *Ecol. Monogr.* **78**, 591–613 (2008).
15. IUCN. The IUCN Red List of Threatened Species. Red List version 2012. <https://www.iucnredlist.org>. Downloaded on 02 October 2014. (2012).
16. Garshelis, D. L. Variation in Ursid life histories: Is there an outlier? in *Giant Pandas*:

Biology and Conservation (eds. Lindburg, D. & Baragona, K.) 53–73 (University of California Press, Berkeley and Los Angeles, California, 2004).

17. Christiansen, P. Feeding ecology and morphology of the upper canines in bears (carnivora: Ursidae). *J. Morphol.* **269**, 896–908 (2008).
18. Mattson, D. J. Diet and Morphology of Extant and Recently Extinct Northern Bears. *Ursus* **10**, 479–496 (1998).
19. Sacco, T. & Van Valkenburgh, B. Ecomorphological indicators of feeding behaviour in the bears (Carnivora: Ursidae). *J. Zool.* **263**, 41–54 (2004).
20. Joshi, A. R., Garshelis, D. L. & Smith, J. L. D. Seasonal and Habitat-Related Diets of Sloth Bears in Nepal. *J. Mammal.* **78**, 584–597 (1997).
21. Wong, S. T., Servheen, C. & Ambu, L. Food habits of Malayan sun bears in lowland tropical forest of Borneo. *Ursus* **13**, 127–136 (2002).
22. Christiansen, P. Evolutionary implications of bite mechanics and feeding ecology in bears. *J. Zool.* (2007).
23. Derocher, A., Andriashek, D. & Stirling, I. Terrestrial foraging by polar bears during the ice-free period in Western Hudson Bay. *Arctic* **46**, 251–254 (1993).
24. Reid, D. G., Jinchu, H., Sai, D., Wei, W. & Yan, H. Giant Panda *Ailuropoda melanoleuca* behaviour and carrying capacity following a bamboo die-off. *Biol. Conserv.* **49**, 85–104 (1989).
25. Schaller, G. B. *et al.* The Feeding Ecology of Giant Pandas and Asiatic Black Bears in the Tangjiahe Reserve, China. in *Carnivore Behavior, Ecology, and Evolution* (ed. Gittleman, J. L.) 212–241 (Springer US, Boston, MA, 1989). doi:10.1007/978-1-4613-0855-3.
26. Matsubayashi, J. *et al.* Major decline in marine and terrestrial animal consumption by brown bears (*Ursus arctos*). *Sci. Rep.* **5**, 9203 (2015).
27. Nie, Y. *et al.* Giant Pandas Are Macronutritional Carnivores. *Curr. Biol.* **29**, 1677-1682.e2 (2019).
28. Mikkelsen, A. J. *et al.* Testing foraging optimization models in brown bears: Time for a paradigm shift in nutritional ecology? *Ecology* e4228 (2023) doi:10.1002/ecy.4228.
29. Robbins, C. T. *et al.* Ursids evolved early and continuously to be low-protein macronutrient omnivores. *Sci. Rep.* **12**, 15251 (2022).
30. Allen, M. L., Elbroch, L. M., Wilmers, C. C. & Wittmer, H. U. The Comparative Effects of Large Carnivores on the Acquisition of Carrion by Scavengers. *Am. Nat.* **185**, 822–833 (2015).

31. Allen, M. L., Elbroch, L. M. & Wittmer, H. U. Can't bear the competition: Energetic losses from kleptoparasitism by a dominant scavenger may alter foraging behaviors of an apex predator. *Basic Appl. Ecol.* **51**, 1–10 (2021).
32. Elbroch, L. M., Lendrum, P. E., Allen, M. L. & Wittmer, H. U. Nowhere to hide: pumas, black bears, and competition refuges. *Behav. Ecol.* **26**, 247–254 (2015).
33. Ordiz, A. *et al.* Individual Variation in Predatory Behavior, Scavenging and Seasonal Prey Availability as Potential Drivers of Coexistence between Wolves and Bears. *Diversity* **12**, 356 (2020).
34. Prugh, L. R. & Sivy, K. J. Enemies with benefits: integrating positive and negative interactions among terrestrial carnivores. *Ecol. Lett.* **23**, 902–918 (2020).
35. Tallian, A. *et al.* Competition between apex predators? Brown bears decrease wolf kill rate on two continents. *Proc. R. Soc. B Biol. Sci.* **284**, 20162368 (2017).
36. Tallian, A. *et al.* Of wolves and bears: Seasonal drivers of interference and exploitation competition between apex predators. *Ecol. Monogr.* **92**, e1498 (2022).
37. Palomares, F. & Caro, T. M. Interspecific Killing among Mammalian Carnivores. *Am. Nat.* **153**, 492–508 (1999).
38. Seryodkin, I. V., Miquelle, D. G., Goodrich, J. M., Kostyria, A. V. & Petrunenko, Y. K. Interspecific Relationships between the Amur Tiger (*Panthera tigris altaica*) and Brown (*Ursus arctos*) and Asiatic Black (*Ursus thibetanus*) Bears. *Biol. Bull.* **45**, 853–864 (2018).
39. Sharp, T. R., Garshelis, D. L. & Larson, W. A most aggressive bear: Safari videos document sloth bear defense against tiger predation. *Ecol. Evol.* **14**, e11524 (2024).
40. Elbroch, L. M. & Kusler, A. Are pumas subordinate carnivores, and does it matter? *PeerJ* **6**, e4293 (2018).
41. Paquet, P. C. & Carbyn, L. N. Wolves, *Canis lupus*, killing denning Black Bears, *Ursus americanus*, in the Riding Mountain National Park Area. *Can. Field-Nat.* **100**, 371–372 (1986).
42. Gunther, K. A. & Smith, D. W. Interactions between wolves and female grizzly bears with cubs in Yellowstone National Park. *Ursus* **15**, 232–238 (2004).
43. Schoeninger, M. J. & DeNiro, M. J. Nitrogen and carbon isotopic composition of bone collagen from marine and terrestrial animals. *Geochim. Cosmochim. Acta* **48**, 625–639 (1984).